

# The influence of residential wood combustion on the concentrations of PM2.5 in four Nordic cities

Kukkonen Jaakko[1], Susana López-Aparicio[2], David Segersson[3], Camilla Geels[4], Leena Kangas[1], Mari Kauhaniemi[1], Androniki Maragkidou[1], Anne Jensen[4], Timo Assmuth[5], Ari Karppinen[1], Mikhail Sofiev[1], Heidi Hellen[1], Kari Riikonen[1], Juha Nikmo[1], Anu Kousa[6], Jarkko V. Niemi[6], Niko Karvosenoja[5], Gabriela Sousa Santos[2], Ingrid Sundvor[7], Ulas Im[4], Jesper H. Christensen[4], Ole-Kenneth Nielsen[4], Marlene S. Plejdrup[4], Jacob Klenø Nøjgaard[4], Gunnar Omstedt[3], Camilla Andersson[3], Bertil Forsberg[8], Jørgen Brandt[4]

[1] Finnish Meteorological Institute, Erik Palmenin aukio 1, P.O. Box 503, 00101, Helsinki, Finland
[2] Norwegian Institute for Air Research, Instituttveien 18, P.O. Box 100, 2027 Kjeller, Norway
[3] Swedish Meteorological and Hydrological Institute, SE-60176 Norrköping, Sweden
[4] Aarhus University, Frederiksborgvej 399, 4000 Roskilde, Denmark
[5] Finnish Environment Institute, Latokartanonkaari 11, FI-00790 Helsinki
[6] Helsinki Region Environmental Services Authority, Ilmalantori 1, FI-00240 Helsinki
[7] Institute of Transport Economics, Gaustadalléen 21, 0349 Oslo, Norway
[8] Umeå University, 901 87 Umeå, Sweden

Correspondence: Jaakko Kukkonen (jaakko.kukkonen@fmi.fi)

**Abstract.** Residential wood combustion (RWC) is an important contributor to air quality in numerous regions worldwide. This study is the first extensive evaluation of the influence of RWC on ambient air quality in several Nordic cities. We have analyzed the emissions and concentrations of $PM_{2.5}$ in cities within four Nordic countries: the metropolitan areas of Copenhagen, Oslo and Helsinki, and Umeå. We have evaluated the emissions for the relevant urban source categories and modelled atmospheric dispersion on regional and urban scales. The emission inventories for RWC were based on local surveys, the amount of wood combusted, combustion technologies and other relevant factors. The accuracy of the predicted concentrations was evaluated based on urban concentration measurements. The predicted annual average concentrations ranged spatially from 4 to 7 µg/m³ (2011), from 6 to 10 µg/m³ (2013), from 4 to more than 13 µg/m³ (2013) and from 9 to more than 13 µg/m³ (2014), in Umeå, Helsinki, Oslo and Copenhagen, respectively. The higher concentrations in Copenhagen were mainly caused by the higher long-range transported background. The annual average fractions of $PM_{2.5}$ concentrations attributed to RWC within the considered urban regions ranged spatially from 0 to 15 %, from 0 to 20 %, from 8 to 30 % and from 0 to 60 % in Helsinki, Copenhagen, Umeå and Oslo, respectively. In particular, the contributions of RWC in central Oslo were larger than 40 % as annual averages. In Oslo, wood combustion was used mainly for the heating of larger blocks of flats. On the contrary, in Helsinki, RWC was solely used in smaller detached houses. In Copenhagen and Helsinki, the highest fractions occurred outside the city center in the suburban areas. In Umeå, the highest fractions occurred both in the city centre and its surroundings. Stricter and more efficient emission regulations should be set in the Nordic countries with respect to RWC, especially in urban areas, for the protection of human health.



# 1. Introduction

Wood combustion or other kinds of biomass for residential heating is a significant source of atmospheric pollution. Biomass combustion has been found to contribute significantly to particulate matter emissions in numerous countries worldwide (e.g., Karagulian et al., 2015, Butt et al., 2016 and Vicente and Alves, 2018). In Europe, EU has promoted the use of biomass for energy production, in order to replace fossil fuels and to reduce greenhouse gas emissions. In many European countries,

national energy policies have also favoured an increased use of biomass, with the intention to promote the use of renewable and domestic energy sources, improve energy security, and decrease the carbon footprint. The use of biomass as an energy source has also commonly beneficial impacts on both national employment, and on the economic development of rural areas.

For simplicity, we mainly use in this article the term 'RWC' (RWC). We define this term to include RWC of various wood products. The concept of small-scale refers here to either detached residential houses, row (terraced) houses, or moderately-sized blocks of flats. Clearly, the burning of oil, coal or municipal waste in residential appliances is outside the scope of RWC. The term 'small-scale combustion' (SSC) has also been used in the literature to refer to combustion in stationary small-scale

appliances. Such appliances can be used, e.g., at homes, in small and medium-scale industry and in heat and energy production. However, this definition does not include small-scale combustion in traffic. Clearly, the concept SSC is more comprehensive, and includes more fuels and sources compared with RWC.

Regarding RWC globally, Vicente and Alves (2018) presented a global scale overview of particulate emissions from residential biomass combustion. They presented and analyzed particulate matter emission factors that have been reported in the literature worldwide. They also discussed the suitability of various organic markers for the source apportionment of residential biomass combustion. According to their review, residential fuel burning has been estimated to be responsible for substantial shares of

particulate matter concentrations in Africa (34 %), in Central and Eastern Europe (32 %), Northwestern Europe (22 %), the Southern China region (21 %), South Eastern Asia (19 %), and India (16 %).

Karagulian et al. (2015) presented a global scale review of the source contributions to the ambient air $PM_{2.5}$ concentrations. According to this review, 25 % of urban ambient air pollution from $PM_{2.5}$ is

contributed to traffic, 15% to industrial activities, 20 % to domestic fuel burning, 22% to unspecified anthropogenic sources, and 18% to natural dust and salt. On the other hand, regarding European regions, viz. Northwestern, Western, Central and Eastern, and Southwestern Europe, they reported that domestic wood burning was responsible for 22 %, 15 %, 32 % and 12 % of the concentrations, respectively.


Two studies for major cities in the UK indicated that the contributions of RWC to particulate matter were clearly lower than those observed for Nordic cities and part of the cities in continental Europe (Fuller et al., 2014, Harrison et al., 2012). Fuller et al. (2014) reported that RWC contributed by 9% to ambient $PM_{10}$ in London in 2010. Harrison et al. (2012) reported RWC contributions below 1% of

ambient $PM_{2.5}$ concentrations in London and Birmingham.

Butt et al. (2016) evaluated the impacts of residential combustion emissions on atmospheric aerosol, human health and climate. They used a global aerosol microphysics model to estimate the impacts for the year 2000. According to their computations, the largest contributions of residential emissions to





PM$_{2.5}$ concentrations occurred in East and South Asia, and Eastern Europe. The global annual excess adult premature mortality was estimated to be 308 000 for the residential emissions. In Europe and North America, 29 000 premature deaths have been estimated to be ascribed annually to residential biomass burning (Chafe et al., 2015).

Cordell et al. (2016) evaluated the impacts of biomass burning in the UK, the Netherlands, Belgium and France. The contributions of biomass burning were quantified using levoglucosan measurements from PM$_{10}$ aerosol filters. Their findings indicated that the contribution of biomass combustion to PM$_{10}$ concentrations was the largest in November and March; more specifically, during the winter it ranged from 2.7 % to 11.6 %. Submicron particulate matter was measured by aerosol mass spectrometers in 13
short-term campaigns across the greater Alpine region from 2002–2009, where wood burning emissions accounted for 17-49 % of organic aerosol in winter (Lanz et al., 2010). Yttri et al. (2019) analyzed the carbonaceous particle fraction at nine European locations during winter, spring and autumn. The contribution of RWC was substantial, accounting for 30-50% of the total carbon in particles at most sites.

Markers of processes and abundant sources of particles including fossil carbon, aging of aerosols, wood combustion and Primary Biological Aerosols Particles (PBAP) were apportioned based on measurements during a summer campaign at four Nordic rural background sites in 2009 (Yttri et al., 2011). In late summer, biomass burning was present but contributed only by 3-7 % to the carbonaceous aerosol, whereas PBAP accounted for 20-32%. According to a study conducted by Hedberg et al.
(2006), RWC in a small city in Northern Sweden corresponded to 70 % of the fine particle mass (measured in the period Jan 15 to March 9, 2002). Molnár and Sallsten (2013) found an increment caused by RWC of 0.9 µg/m$^3$ in a Swedish village during days with prevainling subzero mean temperatures.

     In another study carried out by Glasius et al. (2006), the influence of RWC was studied in a small
Danish rural village. Their results showed that PM$_{2.5}$ concentrations in the village were ca. 4 µg/m$^3$ higher than at a nearby background monitoring site during the winter period. Their findings regarding the observation of high PM2.5 concentrations during the evening and the night was consistent with a local heating source. However, considerable variation in source strengths have been observed within identical seasons from year to year. In a later study, RWC was analyzed in a similar village and season
in the same region (Glasius et al., 2008). The local contribution of RWC to PM$_{2.5}$ averaged 1.2 µg/m$^3$, which corresponds to 10% of ambient PM2.5. Local RWC contributions up to 2.6 µg/m$^3$ were measured in the evening, whereas these contributions were clearly lower during daytime.

     Saarnio et al. (2012) measured the concentrations of levoglucosan and other monosaccharide anhydrides in the Helsinki Metropolitan Area (HMA), to estimate the impact of wood combustion on
PM$_{2.5}$ concentrations. They evaluated that the average contributions of RWC to ambient PM$_{2.5}$ concentrations ranged from 18% to 29% at two urban sites and from 31% to 66% at two suburban sites during various periods within the colder half-year. Regionally distributed wood combustion particles contributed over the whole HMA, whereas particles from local wood combustion sources raised especially the concentrations at suburban sites. Hellén et al. (2015) also observed that the local
emissions from residential wood combustion caused high benzo(a)pyrene (BaP) and levoglucosan





concentrations in the HMA. The BaP concentrations exceeded the European Union target value for the annual average concentrations ($1\,\mathrm{ng\,m^{-3}}$) in certain suburban detached-house areas.

Some studies have addressed specifically particulate carbonaceous matter from wood burning. For example, Genberg et al. (2011) found that the contribution of biomass combustion to carbonaceous aerosol was 32 % during winter at a background station in southern Sweden, based on weekly samples collected for one year. More than 80% of the total carbon was attributed to anthropogenic sources in both Norwegian urban and rural background environments (Yttri et al., 2011). They also reported that fossil-fuel sources were the major anthropogenic contributors during summer, whereas wood burning for residential heating was the major contributor during winter. Szidat et al. (2009) investigated particulate matter at an urban and a rural site in Gothenburg, Sweden, in February and March of 2005. The authors reported that the relative contribution of elemental carbon (EC) from wood burning was more than three times higher at the rural site, compared to the total EC in the urban site. However, the total EC concentration was similar at both sites.

Helin et al. (2018) estimated that the mean contributions of RWC to black carbon (BC) concentrations in the HMA in winter (from December 2015 to February 2016) were clearly higher at two suburban detached house area sites (46%), compared with the corresponding contributions at an urban street canyon site (17%). In the detached house areas, the highest BC concentrations were detected diurnally in the evenings, due to RWC. Aurela et al. (2011) estimated that the mean ratio of organic aerosol originated from biomass burning (BBOA) to the total organic aerosol was fairly high (27–30%) at two suburban detached house areas in the HMA in winter. Especially during cold periods, BBOA contributions up to a half of the total organic aerosol were observed.

Brandt et al. (2013) assessed the contribution of ten major emission sectors in Europe to the health-cost externalities of air pollution. Based on emissions for 2000 and the Economic Valuation of Air pollution (EVA) system, they estimated that non-industrial combustion (dominated by RWC) contributed to ca. 10 % of the total health costs due to air pollution in Europe. In a more recent study using the same model, it was found that RWC within Denmark is associated annually with ca. 500 premature deaths in Denmark (Brandt et al., 2016). Several studies reported that existing regulation, primarily affecting new stoves, effectively reduce emissions from RWC in Europe and North America. Bjørner et al. (2018) calculated net welfare gains regarding the use of wood stoves in Denmark by an integrated assessment model. They concluded that substantial welfare gains could be obtained by imposing a geographically differentiated regulation of existing stoves un Denmark. The results indicated that similar effects could be obtained in the rest of Europe and North America.

The overarching aim of this article is to evaluate the influence of RWC within urban regions on air quality in four Nordic cities, viz. Copenhagen, Helsinki, Oslo and Umeå. The more specific objectives include, first, to present and inter-compare the methodologies for evaluating the emissions and dispersion of fine particulate matter originated from RWC in four Nordic cities. Second, we aim to compare the predicted concentrations with the available relevant air quality measurements. Third, we intend to present and analyze numerical results on the $PM_{2.5}$ concentrations. In particular, we will quantify the influence of RWC in urban regions on the $PM_{2.5}$ concentrations in the selected cities. We will also report and evaluate the current regulations regarding the emissions and concentrations from RWC. This article presents for the first time a systematic assessment of the influences of RWC on air quality in several Nordic cities.



## 2. Methods

This study focuses on three Nordic capital regions: Oslo, Helsinki and Copenhagen, and one smaller city including its neighboring area, Umeå. Our aim was to investigate greater capital or urban areas, instead of solely the areas of the cities. For instance, we address the Helsinki Metropolitan Area which contains four separate cities. However, for simplicity, we chose to refer in the following to the capital regions simply as Oslo, Helsinki and Copenhagen.

Umeå was selected instead of the Swedish capital, due to lack of detailed information on the influence of RWC in Stockholm. This article presents the results for one year for each city. The target years are 2011 for Umeå, 2013 for Helsinki and Oslo, and 2014 for Copenhagen.

We have addressed the contributions of RWC originated from sources within the target urban regions. Clearly, a fraction of the regional background is also originated from RWC that is located outside the considered urban regions.

### 2.1 The considered cities, regions and measurement networks

The locations of the selected cities and the domains are presented in Fig. 1. The considered domain sizes were selected mainly based on the sizes of the cities and their surrounding metropolitan areas; the domain is therefore largest for Copenhagen and smallest for Umeå.

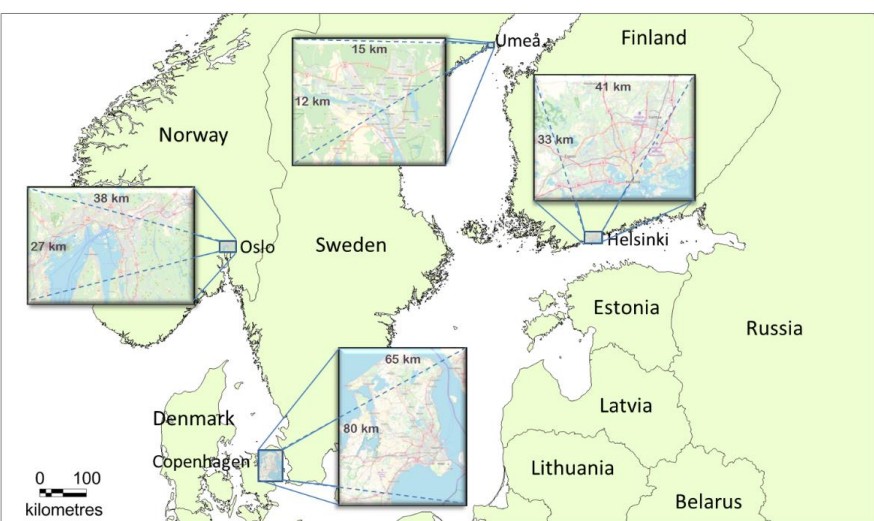

Fig.1. The locations of the selected cities and domains. The physical sizes of the domains have been indicated in the inserted smaller maps.





The geographical locations and the air quality measurement stations addressed in this study are
presented in Figs. 2a-d. All the considered cities are located either on the coast or in the immediate
vicinity of the coast of the Baltic Sea.

Figs. 2a-d. The geographical locations of the cities and the air quality measurement stations for (a)
Umeå, (b) Helsinki, (c) Oslo and (d) Copenhagen. The panels represent the locations of the stations in
2011 for Umeå, 2013 for Helsinki and Oslo, and 2014 for Copenhagen, respectively. The most densely
populated central areas of Helsinki, Oslo and Copenhagen have been shown with light brown color.
Notation for the stations in Oslo: 1: Sofienbergparken; 2: Grønland; 3: Skøyen; 4: Bekkestua; 5:
Vigernes; 6: Alnabru; 7: Rv4, Aker sykehus; 8: Manglerud; 9: Kirkeveien; 10: Bygdøy Alle; 11:





Hjortnes; 12: Smestad; 13: Eilif Dues vei. © OpenStreetMap contributors 2019. Distributed under a Creative Commons BY-SA License.

### 2.1.1 Geographical regions and climates of the cities

Umeå is the largest city and the capital of the county of Västerbotten in northern Sweden. The domain addressed in this study includes the city and its surrounding areas. The terrain in the county rises from the gulf through a forested upland zone, and culminates in mountains near the Norwegian frontier. The city of Umeå is a medium sized Swedish municipality with 0.12 million inhabitants (all the population counts in this section are for 2018). The Ume River flows through the middle of the city and enters a

bay in the Baltic Sea, at a distance of approximately five kilometers downstream of the city border. The yearly average temperature is 2.7 °C; average monthly temperatures vary from -7 °C in February to +16 °C in July (all the average temperatures presented in this section are based on the standard period of 1961-1990).

The Helsinki Metropolitan Area includes four cities: Helsinki, Espoo, Vantaa and Kauniainen. The

225 total population in the area was approximately 1.1 million, whereas the population of Helsinki was about 0.64 million in 2018. The Helsinki Metropolitan Area is situated on a fairly flat coastal area. The annual mean temperature in Helsinki is 5.9 °C and the average monthly temperatures vary from -5 °C in February to +18 °C in July.

The city of Oslo and the Greater Oslo Region are situated at the northernmost end of a fjord and

230 surrounded by hills that have heights of approximately 500 m above the sea level. The total population in Oslo was approximately 0.63 million in 2018, whereas the metropolitan area had a population of 1.7 million. Oslo has a humid climate; average monthly temperatures vary from -5 °C in January to +17 °C in July.

Copenhagen is situated on the flat eastern coast of the island Zealand; it is separated from Sweden by

235 the narrow Øresund strait. The total population in the urban area of Copenhagen is approximately 1.6 million, whereas the city of Copenhagen has a population of 0.78 million. For the greater area of Copenhagen, the average monthly temperatures range from 0 °C in February to +16 °C in July.

### 2.1.2 Concentration measurement networks

**Concentration measurements for Umeå**

For Umeå, we have addressed both long-term measurements and the results of a measurement campaign. The long-term measurements were conducted from 2006 to 2011 at two sites in the city of Umeå (Västra Esplanaden and Biblioteket). The site of Västra Esplanaden is classified as an urban traffic site; it is a roadside station located in a street canyon with relatively dense traffic. The site of

245 Biblioteket is classified as an urban background site; it is located on a rooftop in central Umeå. The long-term measurements were conducted using TEOM 1400A (Thermo Fisher Scientific, Waltham, MA, USA).



A monitoring campaign was also carried out to evaluate the performance of the modelling approach (Omstedt et al. 2014). The measurements were carried out in the villages of Sävar, Vännäs and Vännäsby, situated in the vicinity of Umeå, and at Tavleliden, located in the southernmost outskirts of the city. The stations of Sävar, Vännäs, Vännäsby and Tavleliden are classified as residential sites.

All monitoring campaign measurements of PM$_{2.5}$ were carried out using filter collection. For Sävar and Vännäsby, the filters were changed on a daily basis and for Tavleliden and Vännäs, on weekly intervals. The analysis of the filters was gravimetric (weighting before and after measurements under standardized conditions).

### Concentration measurements for Helsinki

For this study, we have selected three measurement stations that mainly represent the influence of RWC in residential areas (Vartiokylä, Tapanila and Kauniainen) and three stations that represent either pollution originated from vehicular traffic in the center of Helsinki (Mannerheimintie) or at smaller regional urban centres within the Helsinki Metropolitan Area (Leppävaara and Tikkurila). In addition, we have selected two stations that represent urban (Kallio2) and regional background (Luukki). All the PM$_{2.5}$ monitors were equivalent reference instruments (i.e., TEOM 1400AB, SHARP 5030, FH 62 I-R and Grimm 180).

### Concentration measurements for Oslo

All the available monitoring stations in Oslo in 2013 were classified as either urban or suburban traffic, or urban background. There were no stations originally designed to measure the influence of residential combustion; however, several stations were influenced by pollution from RWC.

At all the considered monitoring stations in Oslo, PM$_{2.5}$ is measured by continuous monitors and logged with a time resolution of 1 hour. All monitors are equivalent reference instruments (i.e., TEOM 1400A, TEOM1405DF-FDMS and Grimm-EDM180).

### Concentration measurements for Copenhagen

The Danish Air Quality Monitoring Network includes five measuring sites in close vicinity of Copenhagen. There are three sites in central Copenhagen: two street sites and one urban background site. We have also used data measured at a suburban site of Hvidovre, located outside of Copenhagen, and at a regional background site in a rural area at Risø. The PM$_{2.5}$ observations were performed using the Low Volume Sampling reference method.

### 2.2 Emission inventories for the target cities

The assessment of emissions located within the target cities is addressed in this section. The regional and continental scale emissions are discussed in the context of regional dispersion modelling. We present first an overview and summary of the emission modelling both for RWC and for all the other urban sources. More detailed descriptions of the assessment of RWC emissions are presented in the following section.

### 2.2.1 Overview of the emission inventories





An overview of the emission inventories regarding RWC is presented in Table 1. In all the cities, the assessment of the emissions from RWC was based on (i) surveys regarding the amounts and use of wood stoves, boilers and other relevant appliances, (ii) national or literature based emission factors, and (iii) the assessments of the spatial distribution of the emissions. In case of Umeå, Helsinki and Copenhagen, also various national or local register data been used.


Table 1. Assessment of the emissions of $PM_{2.5}$ originated from RWC, and their spatial resolution in the target cities.

| | Umeå | Helsinki | Oslo | Copenhagen |
|---|---|---|---|---|
| **Data and information sources ragarding the use of wood for combustion, and on the combusion appliances** | (i) Survey on the amounts of wood stoves and boilers, and the habits of wood combustion. (ii) Register data gathered by chimney sweepers. | (i) Survey concerning the amount of wood combusted, types and amounts of fireplaces, habits of wood combustion, for detached and semi-detached houses. (ii) Regional basic register for dwellings | (i) Survey regarding the amount and temporal variability of wood combusted by Statistics Norway. | (i) Survey on the unit consumption and age for different types of residences. (ii) Register data on the location of the appliances from the chimney sweepers. (iii) Danish energy statistics and building and dwelling register. (iv) The spatial distribution is evaluated by the SPREAD model (Plejdrup et al., 2016) |
| **Assessment of emission factors** | Combination of results from national measurement programmes and available literature (Omstedt et al., 2014) | Combination of results from national measurement programmes and available literature. (Kaski et al., 2016, Savolahti et al., 2016) | National measurements reported by Haakonsen and Kvingedal (2001). | Combination of results from the EMEP/EEA Guidebook (EMEP/EEA, 2016) and national measurements. |
| **Spatial resolution of the predicted emissions of $PM_{2.5}$** | Appliances were treated as point sources. | 100 x 100 m$^2$ | 1 x 1 km$^2$ | 1 x 1 km$^2$ |
| **Basis for spatial allocation of emissions, i.e., gridding** | Geocoded addresses of combustion appliances, based on a survey and chimney sweeper register | Average wood use for houses with different primary heating methods. Location of the houses from local building and dwelling register. | The amount of wood consumed in the districts in Oslo, based on a survey carried out by Statistics Norway. | Average wood consumption in different types of houses and location of the appliances based on chimney sweepers register. Location of the houses from Danish building and dwelling register. |
| **Basis for temporal allocation of emissions** | Measured local contributions of the concentrations of $PM_{2.5}$ as a proxy variable | Information gathered in questionnaires (Kaski et al., 2016) | Based on a survey carried out by Statistics Norway | Based on the SPREAD model |





Information on the combusted wood is subsequently combined with the corresponding emission factors. The assessment of emission factors has been based on either on national measurements (Oslo), or a combination of national measurements and results from the available literature (Umeå, Helsinki and Copenhagen). All measurements that were used for the assessment of emission factors were based on methodologies using cooled flue gases and dilution chambers.

Clearly, the RWC emissions are dependent on the temporal variation of the meteorological conditions, especially on the ambient temperature. In case of Oslo, the variation of emissions on the ambient temperature has also been taken into account, based on measured weekly average ambient temperatures. In the inventory for Umeå, individual sources were treated separately. For the other cities, the computed emissions have been gridded on various spatial resolutions, from 100 x 100 m$^2$

(Helsinki) to 1 x 1 km$^2$ (Oslo and Copenhagen).

An overview of the emission inventories for the other relevant source categories is presented in Table 2. The vehicular traffic exhaust emissions have been included for all the cities. The suspension emissions originated from vehicular traffic have been included for Umeå, Helsinki and Oslo. The emissions from shipping have been included for Umeå, Oslo and Copenhagen. In case of Helsinki,

Kukkonen et al. (2018) presented a detailed analysis regarding the contribution of shipping on the PM2.5 concentrations, based on computations for a three-year period. They found that the contribution of shipping, including harbour activities, to the ambient air PM2.5 concentrations varied from 10 to 20 % near major harbours to a negligible contribution in most other parts of the metropolitan area.

However, the emission inventories for other source categories except for RWC were not the main focus

of this article. Their more detailed descriptions have therefore been presented in Appendix A.

### 2.2.2 Detailed descriptions of the assessment of emissions from RWC

For the estimation of the emissions of wood combustion, one needs to know numerous factors, including (i) the spatial distributions of the various categories of buildings using wood combustion, (ii) the amounts and distribution of firewood used, (iii) the shares of primary and secondary heating

sources, (iv) the amounts of wood used and the numbers of boilers, stoves, fireplaces, sauna stoves and other heating devices, and (v) the emission factors for the different types of heating devices (Kukkonen et al., 2018).

The information on the use of wood and the heating device technologies is mostly based on surveys. Moreover, in case the survey year and the study year are not the same, the information on the changes

of technologies and fuels in time is also needed. There are also other factors that may have a substantial influence on the assessment of RWC emissions, which are commonly estimated in a simplified manner, or even neglected in evaluating the emissions of RWC (e.g., Savolahti et al., 2016). These include (i) the compositions of wood fuels, e.g., their humidity, the tree species and the pre-processing and storage of wood, and (ii) the variations of the habits and procedures of combustion (Kukkonen et al., 2018).

For these reasons, the uncertainties in the RWC emission estimates of PM$_{2.5}$ are commonly relatively higher than those for most other major emission source sectors (e.g., Karvosenoja et al., 2008).



Table 2. Assessment of the traffic flows and emissions from vehicular traffic and other source categories, except for RWC, in the target cities.


| | | Umeå | Helsinki | Oslo | Copenhagen |
|---|---|---|---|---|---|
| **Vehicular traffic flows and emissions** | **Vehicular traffic flows** | Traffic flow model EMME/2 and measured data | Traffic flow model EMME/2 and measured data | Traffic flow model RTM23+ | National GIS-based road network and traffic database. The spatial distribution is done by the SPREAD model |
| | **Vehicular exhaust emissions** | Emission factors by Hausberger (2009) | The LIPASTO emission model | NILUs traffic emission model | Danish area: the SPREAD emission model |
| | **Vehicular suspension emissions** | Resuspension model by Omstedt (2006) | The FORE traffic suspension emission model (Kauhaniemi et al., 2011) | The NORTRIP traffic suspension emission model (Denby et al., 2013) | Not included |
| **Shipping emissions** | | Modelled using SHIPAIR (Segersson, 2013) | Not included in the modelling | Based on López-Aparicio et al. (2017b) and US EPA (2009) | An updated version of AIS based inventory for Denmark (Olesen et al., 2009). |
| **Other sources** | | National compilation of emissions originated from off-road machinery and major point sources in Sweden. | Not included in the modelling | Industrial emissions and emissions from off-road mobile combustion | Fugitive emissions from fuels, emissions from industrial processes, agriculture and waste modelled by SPREAD |

## The assessment of emissions from RWC for Umeå

A survey regarding the habits of wood consumption and combustion was carried out in four areas in 2013, which included a recently constructed suburb and three small towns. The survey included also an air quality monitoring campaign. Based on the register data gathered by the chimney sweepers, we selected a representative sample of 178 houses with a stove or a boiler. A total of 176 houses were willing to participate to the survey; these households were subsequently visited. The residents were interviewed using a form with questions mainly regarding the type of stove or boiler, the principal type

of heating, biofuel consumption, biofuel type, combustion habits and the actions to reduce energy consumption.

A bottom-up inventory was made on the amounts of wood stoves and boilers, partially based on the above-mentioned survey. The inventory used also register data that had been gathered by the local chimney sweepers. The inventory was compiled in the Västerbotten county in 2009. This dataset

included information on the types of equipment, such as boilers (wood or oil), stoves, pellets boiler and





open fireplaces, and their geocoded addresses. A total of more than 54 thousand appliances were identified within the county. About 23 % of them were wood boilers, 10 % pellet boilers, 64 % stoves and 3 % oil boilers.

We evaluated the amounts of combusted wood and the emission factors based on dilution chamber experiments by Omstedt et al. (2014). Separate emission factors were used for (i) wood, (ii) pellet and (iii) oil fueled boilers, (iv) fireplaces and stoves, and (v) summer houses and cottages.

The temporal variations of the emissions originated from wood combustion were evaluated using the measured local contributions of the concentrations of $PM_{2.5}$ as a proxy variable. The local contributions of the $PM_{2.5}$ concentrations were estimated by subtracting the modelled regional background 360 concentration from the local measurements. All measurement stations used for these estimations were located in areas with a substantial amount of RWC.

**The assessment of emissions from RWC for Helsinki**

Emissions from RWC were based on an emission inventory for the years 2013-2014, including the spatial and temporal variation of emissions. We estimated by using a questionnaire the amount of wood 365 combusted in 12 different fireplace types, and the procedures and habits for the combustion. Its results were applied for all detached and semi-detached houses in the area.

The spatial distribution of the emissions was based on average wood use per combustion appliance type for each main heating method of a house, based on the questionnaires (Kaski et al., 2016). The emissions were allocated to the location of the houses available in local building and dwelling register, 370 and the emissions were allocated to the 100 x 100 $m^2$ grid.

The temporal variation (monthly, weekly, hourly) of emissions was estimated based on the information gathered in questionnaires (Kaski et al., 2016). The temporal variation was estimated separately for three different source categories: heating boilers, sauna stoves, and other fireplaces. However, the information was not sufficient to model quantitatively the influence of meteorological variables on the 375 emissions.

The emission factors for different types of fireplaces were adopted based on the results of national measurement programmes and the literature (Kaski et al., 2016, Savolahti et al., 2016). The spatial distribution of RWC emissions was based on the regional basic register for dwellings, provided by the Helsinki Region Environmental Services Authority; this register contains information on primary 380 heating methods.

**The assessment of emissions from RWC for Oslo**

The RWC emissions were estimated based on a bottom-up approach by using the data of a dedicated survey. The survey was carried out by Statistics Norway; its aim was to assess the use of wood combustion and heating habits in Oslo. The results of the survey include information on the amount of 385 wood consumed in the districts in Oslo, and information on how the wood combustion varies temporally, in terms of weeks, days and hours of the day. Information on the amount of wood combusted was collected based on the survey in terms of the type of technology, i.e., open fireplace, wood stove produced before 1998 and wood stove produced after 1998.



The emissions factors were extracted from Haakonsen and Kvingedal (2001), which was based on a
review of the results from different tests for various fireplaces in Norway. Separate emission factors
were used for conventional wood stoves, certified wood stoves and open fireplaces.

The seasonal variations of emissions was taken into account, by modelling their variation using their
dependency on the ambient temperature, based on observed weekly average ambient temperatures. The
weekly mean temperatures measured at the station of Blindern in 2013 were used in the
parametrization.

**The assessment of emissions from RWC for Copenhagen**

A survey was conducted regarding the unit consumption of wood and age of different types of
residences by the Danish Technological Institute in 2015. A distinction was made between villas,
apartments and allotments, either connected or unconnected to district heating. The survey also
included information on the age of the appliance, distributed into four age categories. For RWC in the
Copenhagen area, detailed data was also used on the location of the appliances based on the chimney
sweeper register data for Copenhagen in 2015.

The assessment of the emissions for the Danish area were based on the SPREAD model. The SPREAD
model is an integrated database system for high-resolution (1 km x 1 km) spatial distribution of
emissions (Plejdrup et al., 2016 and 2018). The SPREAD model includes emission distributions for
each sector in the Danish emission inventory system. In this study, the emission factors included in this
national inventory were used (Nielsen et al., 2018). These were based on emission factors of the
EMEP/EEA Guidebook (EMEP/EEA, 2016) and national measurements.

The emission inventory for RWC was also based on wood consumption information by the Danish
energy statistics. The spatial distribution of RWC emissions was based on the Danish building and
dwelling register, which includes information on building use, and on primary and secondary heating
installations.

**2.3 Atmospheric dispersion modelling for the target cities**

We present first an overview and summary of the dispersion modelling, and second, a more detailed
description of dispersion modelling in the target cities.

**2.3.1 Overview of dispersion modelling**

An overview of the dispersion modelling has been presented in Table 3. The assessment of the regional
background concentrations was based on chemical transport modelling in all the cities, except for
Umeå, for which the assessment of the regional background was based on measured data. For the urban
scale assessments, multiple-source Gaussian modelling systems were used for all the cities. As the
focus on this study was on RWC, the dispersion in street canyons was modelled only for one street
canyon measurement station in Umeå. The spatial resolutions of the modelling of the dispersion
originated from RWC ranged from a couple or a few tens of meters (Oslo, Umeå) to 100 m (Helsinki)
and 1 km (Copenhagen).





Table 3: Atmospheric dispersion modelling and its spatial resolution in the target cities.

| | | Umeå | Helsinki | Oslo | Copenhagen |
|---|---|---|---|---|---|
| **Assessment of regional background concentrations** | | Measured values at a regional background station | Predictions of the regional and global scale chemical transport model SILAM | Predictions of model ensemble, using seven regional-scale chemical transport models | Predictions of the hemispheric chemical transport model DEHM |
| **Urban scale dispersion modelling** | **Residential wood combustion** | Multiple-source Gaussian model DISPERSION | Multiple-source Gaussian model UDM-FMI | Multiple-source Eulerian model EPISODE | The Gaussian plume-in grid model - Urban Background Model (UBM) |
| | **Vehicular traffic for the whole city** | Multiple-source Gaussian model DISPERSION | Roadside dispersion model CAR-FMI | Multiple-source Eulerian model EPISODE, including sub-grid Gaussian line source modelling | The Gaussian plume-in grid model - Urban Background Model (UBM) |
| | **Vehicular traffic in street canyons** | Street canyon dispersion model OSPM was used. | Street canyon modelling (OSPM) is included in the modelling system, but was not used in this study. | Street canyon modelling was not included in the modelling system. | Street canyon modelling (OSPM) was included in the modelling system, but was not used in this study. |
| **Spatial resolution** | | Near the sources 50 x 50 $m^2$, at substantial distances from the sources 3 $km^2$ | Vehicular traffic: from 20 m in the vicinity of traffic sources to 500 m on the outskirts of the area. RWC: 100 x 100 $m^2$ | For the entire modelling domain 20 x 20 $m^2$ | For the entire modelling domain 1 x 1 $km^2$ |


### 2.3.2 Detailed descriptions of dispersion modelling

For each domain, we address first the assessment of the regional background concentrations, and second, the dispersion of urban contributions to concentrations.

**Atmospheric dispersion modelling for Umeå**

The regional background contribution was estimated based on the measured data from two regional background stations (Bredkälen and Vindeln) and on the modelled spatial concentration distributions. The regional background concentrations in 2013 were estimated based on measured values at the regional background station of Bredkälen, adjusted slightly based on regional-scale dispersion model





computations. This station is situated approximately 350 km to the west of Umeå. The regional
background station at Vindeln is situated 50 km north-west of Umeå.

For the year 2013, to account for the influence of concentration gradients between Umeå and the
station of Bredkälen, we have added a contribution of 1.28 $\mu gm^{-3}$ to the measured concentrations at
Bredkälen, based on the computations by Omstedt et al. (2014). Similar yearly adjustments were made
also for years 2006-2011, based on results from the atmospheric chemistry transport model MATCH
and corrections using earlier measurements at the more nearby station Vindeln (Segersson et. al. 2014).

The larger spatial scale meteorological values were extracted from the predictions of the Swedish
version of the numerical weather prediction model HIRLAM, with a horizontal resolution of 22 km.
The finer, mesoscale meteorological data for dispersion modelling was provided by the operational
mesoscale analysis system Mesan (Häggmark et al., 2000), which is based on an optimal interpolation
technique. All available measurements from synoptic and automatic stations, radars and satellites were
analyzed with hourly time resolution on an 11x11 $km^2$ grid across northern Europe. The following
meteorological parameters were used: wind speed and direction at a height of 10 m, ambient
temperature and humidity at a height of 2 m, cloud cover, global radiation and precipitation. Boundary
layer parameters such as friction velocity, sensible heat flux and boundary layer height were calculated
using methods from van Ulden and Holtslag (1985), Holtslag et al. (1995) and Zilitinkevich and
Mironov (1996).

The dispersion of pollutants from RWC and vehicular traffic was modelled using the Gaussian
multiple-source dispersion model DISPERSION (Omstedt, 1988). The DISPERSION model contains a
Gaussian finite length line source dispersion model. For point sources, the DISPERSION model
includes a revised version of the Gaussian OML (Operational Meteorological Air Quality model) point-
source model (Omstedt et al. 2011). For a more detailed description of the model and its evaluation
against experimental data, the reader is referred to Omstedt et al. (2011) and Gidhagen et al. (2013).

The dispersion parameters of the DISPERSION model are continuous functions of boundary-layer
parameters, such as friction velocity, sensible heat flux and boundary layer height. The model also
includes a detailed description of plume rise and building downwash effects. The OML model has
previously been used to investigate the influence of wood combustion on particulate matter
concentrations in residential areas in Denmark (Glasius et al., 2008) and in the northern part of Sweden
(Omstedt et al., 2011). In case of sources described using spatially gridded emissions, a Gaussian
model included in the Airviro air quality management system was applied (SMHI, 2017). Segersson et
al. (2017) have presented a more detailed description of dispersion modelling methodology for other
sources than RWC.

The chimney height for RWC was set to 5 m and the effective plume rise was then evaluated by the
model depending on meteorological conditions. The concentrations were computed on a receptor grid
that was different for the contributions from RWC and vehicular traffic.

The OSPM (Operational Street Pollution Model; Berkovicz, 2000) was used to estimate the
concentrations at the considered street canyon measurement station. The OSPM model was run twice,
both with and without the influence of the surrounding buildings. The difference of these model
computations is a measure for the concentration increment caused by the buildings.





**Atmospheric dispersion modelling for Helsinki**

The regional background concentrations were computed using the SILAM model (Sofiev et al., 2006, 2015) for the European domain. A detailed description of these computations has been presented by Kukkonen et al. (2018). For this study, we selected four grid points of the SILAM computations that were closest to the Helsinki Metropolitan Area (HMA), but outside the urban domain. We then 485 computed an hourly average of the concentration values at these four locations, and used that value as the regional background for all the chemical components of particulate matter, except for mineral dust. In case of mineral dust, we used the lowest hourly value within the four selected points. The latter procedure was adopted to avoid the potential double counting of occasional releases of dust originating from the considered urban area.

The meteorological input variables for the urban scale modelling were based on synoptic weather observations from the stations of Helsinki-Vantaa airport (18 km north of city center) and Harmaja (marine station south of Helsinki), radiation measurements of Helsinki-Vantaa, and sounding observations from Jokioinen (90 km northwest of Helsinki) for the year 2013. Measured meteorological data was analyzed using the meteorological pre-processing model of the Finnish Meteorological 495 Institute (MPP-FMI) adapted for urban environment (Karppinen et al 2000a). The MPP-FMI model is based on the energy budget method of van Ulden and Holtslag (1985), and its output consists of hourly time series of meteorological data needed for the dispersion modelling, including temperature, wind speed, wind direction, Monin-Obukhov length, friction velocity, and boundary layer height. The same meteorological parameters were used for the whole HMA.

For urban dispersion modelling, we used a roadside dispersion model and a multiple source Gaussian model. We did not model dispersion in street canyons.

The urban scale dispersion of vehicular emissions was evaluated with the CAR-FMI model (Contaminants in the Air from a Road – Finnish Meteorological Institute; e.g., Kukkonen et al. 2001). The model is a Gaussian finite line source model, which computes an hourly time-series of the 505 pollutant dispersion. The dispersion parameters are modelled as a function of Monin-Obukhov length, friction velocity, and boundary layer height. The modelling system containing the CAR-FMI model has been evaluated against the measured data of urban measurement networks for gaseous pollutants and particulate matter in the HMA and in London (e.g. Karppinen et al., 2000c, Kousa et al., 2001, Kauhaniemi et al., 2008, Aarnio et al., 2016, Sokhi et al., 2008, Singh et al., 2014), and for gaseous 510 pollutants also against the results of a field measurement campaign and other roadside dispersion models (Kukkonen et al., 2001, Ottl et al., 2001, Levitin et al., 2005).

The dispersion of RWC emissions was evaluated with Urban Dispersion Model of Finnish Meteorological Institute UDM-FMI (Karppinen et al., 2000b). The model is a multiple-source Gaussian dispersion model for various stationary source categories (point, area and volume sources). The 515 modelling system has been evaluated against measurement data of urban measurement networks (e.g., Karppinen et al., 2000b, Kousa et al., 2001).

In this study, the RWC emissions were treated as area sources of the size of 100 m x 100 m. The height of the sources was assumed to be equal to 7.5 m, including the initial plume rise. This altitude was assumed to be the combined average height of detached and semi-detached houses and chimneys in the



area. No chemical reactions or aerosol transformation processes were included in the urban scale computations.

### Atmospheric dispersion modelling for Oslo

The regional background concentrations were extracted from the ensemble reanalysis that comprised of
seven regional-scale chemical transport models (Marécal et al., 2015): CHIMERE, EMEP, EURAD-IM, LOTOS-EUROS, MATCH, MOCAGE and SILAM. Within this ensemble, the models had a common framework in terms of meteorology, chemical boundary conditions and emissions. However, the models have differences in terms of their aerosol representations, chemistry schemes, physical parametrizations, and different implementations for use of the input data.

The meteorological variables used as modelling input were hourly measurements extracted from the data of the meteorological stations in the simulated domain (the stations of Valle Hovin, Blindern, Alna, Tryvannshøgda and Kjeller). All these stations are located within the Oslo municipality, except for the station of Kjeller, which is located at a distance of approximately 25 km northeast. The variables related to wind and atmospheric stability were used as input in a preprocessing diagnostic
wind field model. The hourly wind field data produced by the wind field model were input to the urban scale dispersion modelling.

The atmospheric dispersion modelling was done with the EPISODE model. This model is a combined three-dimensional Eulerian and Lagrangian air pollution dispersion model, which has been developed for urban and local scale applications (Slørdal et al., 2003; Slørdal, 2008). The Eulerian part of the
model consists of a numerical solution of the atmospheric mass conservation equation of the pollutant species in a three-dimensional grid. The Lagrangian part consists of separate subgrid-models for line and point sources. The line source model is an integrated Gaussian type model, whereas the point source model is a Gaussian puff trajectory model. The EPISODE model has been used for a large number of applications, including the assessment of air quality and air pollution control measures in
urban areas (e.g., Sundvor and López-Aparicio, 2014), and in a forecasting systems for seven city regions in Norway.

### Atmospheric dispersion modelling for Copenhagen

The Danish multiscale integrated model system THOR (Brandt et al., 2001, 2003) has for this study
been setup for a domain over Greater Copenhagen. The system combines the Danish Eulerian Hemispheric model (DEHM) and the Urban Background Model (UBM).

The DEHM model (Christensen 1997) is a chemistry-transport model describing the concentration fields of 73 photo-chemical compounds (including NOx, SOx, VOC, NHx, CO, etc.) and nine classes of particulate matter (e.g., $PM_{2.5}$, $PM_{10}$, TSP, seasalt, and fresh and aged black carbon). The regional
model covers the Northern Hemisphere, with higher resolution over Europe (50 km x 50 km), Northern Europe (16.7 km x 16.7 km) and Denmark (5.6 km x 5.6 km). The DEHM model has been extensively evaluated (Brandt et al. 2012; Zare et al. 2014; Solazzo, et al. 2012a, b).





The regional background concentrations were extracted on a 5.6 x 5.6 km$^2$ grid. The meteorological fields were provided by the Weather Research and Forecasting (WRF) Model (Skamarock et al., 2005), using the same domains as the DEHM model. The anthropogenic emissions for the regional modelling were based on a combination of a number of emission inventories including in particular the EMEP emissions for Europe (http://www.ceip.at/webdab_emepdatabase/emissions_emepmodels/). Within the Danish area, the emissions were based on the SPREAD emissions model. Temporal profiles of emissions, depending on the emission type were included.

The Urban Background Model (UBM) is a Gaussian plume model, including a simplified description of photochemical reactions of NO$_x$ and ozone. Themodel was set up for the selected urban domain on a resolution of 1 x 1 km$^2$ and hourly background concentrations were provided by the DEHM model. The UBM model has been used for assessments of air pollution in Denmark, e.g., as part of the Danish AirGis system (Hvidtfeldt et al., 2018, Khan et al., 2018).


### 2.4 Statistical model performance parameters

For simplicity, we have mainly considered two selected statistical model performance parameters: the index of agreement and the fractional bias.

The index of agreement is defined as (Willmott, 1981)

$$IA = 1 - \frac{\sum_{i=1}^{n}(P_i - O_i)^2}{\sum_{i=1}^{n}(|P_i - \bar{O}| + |O_i - \bar{O}|)^2} \tag{1}$$

where n is the number of data points, and P and O refer to predicted and observed pollutant concentrations, respectively. Overbar refers to an average value. Factor-of-two is defined as the fraction of data for which $0.5 \leq P/O \leq 2$.

Fractional bias is given by

$$FB = \frac{2(\bar{P} - \bar{O})}{\bar{P} + \bar{O}} \tag{2}$$

where P and O are the mean values of the predicted and observed values.

## 3. Results

First, the numerical predictions will be evaluated against measured urban scale data regarding the PM$_{2.5}$ concentrations in the four target cities. Second, the predicted emissions originated from RWC will be presented and analyzed. Third, the ambient air concentrations of PM$_{2.5}$, and the contributions from RWC to these concentrations will be presented and discussed. We have also presented an overview of the regulatory frameworks regarding RWC in four Nordic countries in Appendix D.

### 3.1 Evaluation of the predicted concentrations against measured data



The results of the model evaluation are summarised and reviewed in this section. The detailed model evaluation results have been presented in Appendix B.

The ranges of values of two statistical parameters for the daily average concentration values of PM$_{2.5}$ values are presented in Figs. 3a-b, viz. Index of Agreement (IA) and Fractional Bias (FB). The IA is a

measure of the agreement of the measured and predicted time series of concentrations, whereas FB is a measure of the agreement of the average (annual or during several months) values of the concentrations. In case of regional and urban background stations, we have selected one station for each city, whereas for traffic and RWC stations, the range of values is shown by a vertical line, and the value for each station is shown by short horizontal lines.


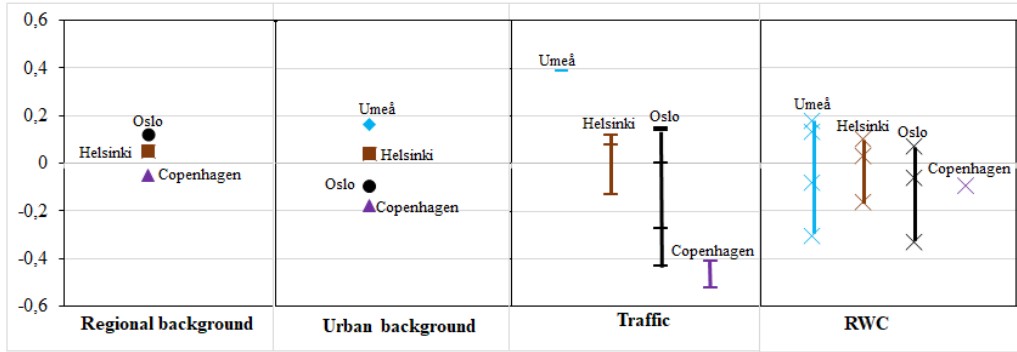

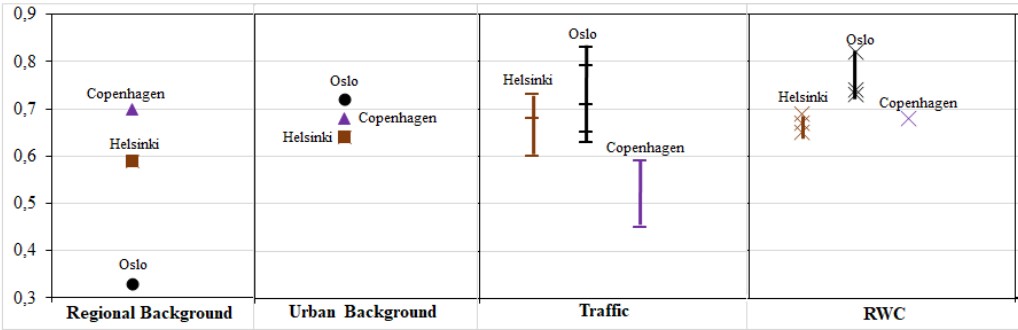

Figs. 3a-b. Values of two statistical model performance measures for the target cities, for various categories of stations. Upper panel presents the fractional biases, the lower panel the index of
agreement. In case of Oslo, we have selected three stations to be representative for RWC (Akerbergveien, Bygdoy Alle and Kirkeveien), although these were officially classified as traffic monitoring stations.

In case of Umeå, the temporal variations of the emissions originated from wood combustion were
evaluated using the measured local contributions of the concentrations of PM$_{2.5}$. It was therefore not





possible to perform an evaluation of the temporal variation of the predicted values for Umeå; the IA values have therefore not been presented for that city.

In case of Oslo, there were no measurement stations that would have been officially nominated by the local authorities as measuring the influence of RWC. We have therefore selected three stations that we
considered to be most influenced by RWC.

The results in Figs. 3a-b facilitate an assessment of model performance in terms of the cities and the categories of the stations. The FB values are reasonably good, considered here as the range from -0.20 to + 0.20, for all the regional and urban background values, and for most of the traffic and RWC stations. However, for some of the traffic and RWC sites, the FB values are substantial, especially for
two traffic stations in Copenhagen (substantial underprediction of the model), one traffic station in Umeå (over-prediction), and two traffic and one RWC station in Oslo (underprediction).

The IA values are also fairly good, considered here as IA > 0.55, in most cases. The agreement of the time series of daily measured and modelled values is relatively worse for the regional background values in Oslo, and for one traffic station in Copenhagen. In particular, the IA values for the traffic
stations are lower for Copenhagen, compared with the corresponding values in Helsinki and Oslo. This is due to the coarser spatial model resolution (1 x 1 km$^2$) in Copenhagen, compared with those in the other three target cities, which tends to result in an underprediction of the local influence of vehicular traffic. A better model performance was obtained in a previous study for the street stations in Copenhagen, when the street pollution model OSPM was used (Khan et al., 2018). For the finer
resolution computations for Helsinki and Oslo, there is no substantial systematic difference between the model performance at traffic stations compared with the corresponding RWC stations.

The measured and predicted annual average concentrations have been summarised in Fig. 4. Both the measured and predicted concentration values are the highest for Copenhagen, caused mainly by the relatively higher regional background contributions, compared with the other three cities. The
concentrations are second highest for Oslo, mainly due to substantial urban contributions.

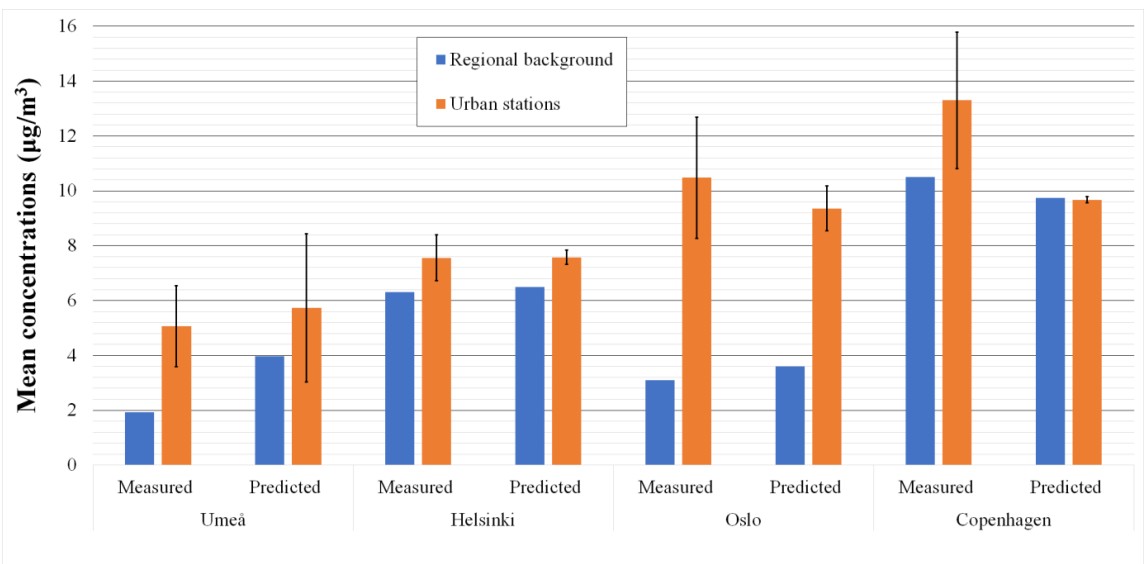

Fig. 4. The measured and predicted long-term average concentrations of PM$_{2.5}$ in the target cities. The contributions of regional background have been shown separately (blue bars). Both the predicted and measured values at the urban stations (brown bars) are averages over all the considered urban measurement stations in each city. The standard deviations of these urban values have been marked with the vertical solid lines.

## 3.2 Emissions of PM$_{2.5}$ originated from RWC

The results of the emission inventories regarding RWC for PM$_{2.5}$ have been presented in Figs. 5a-d.

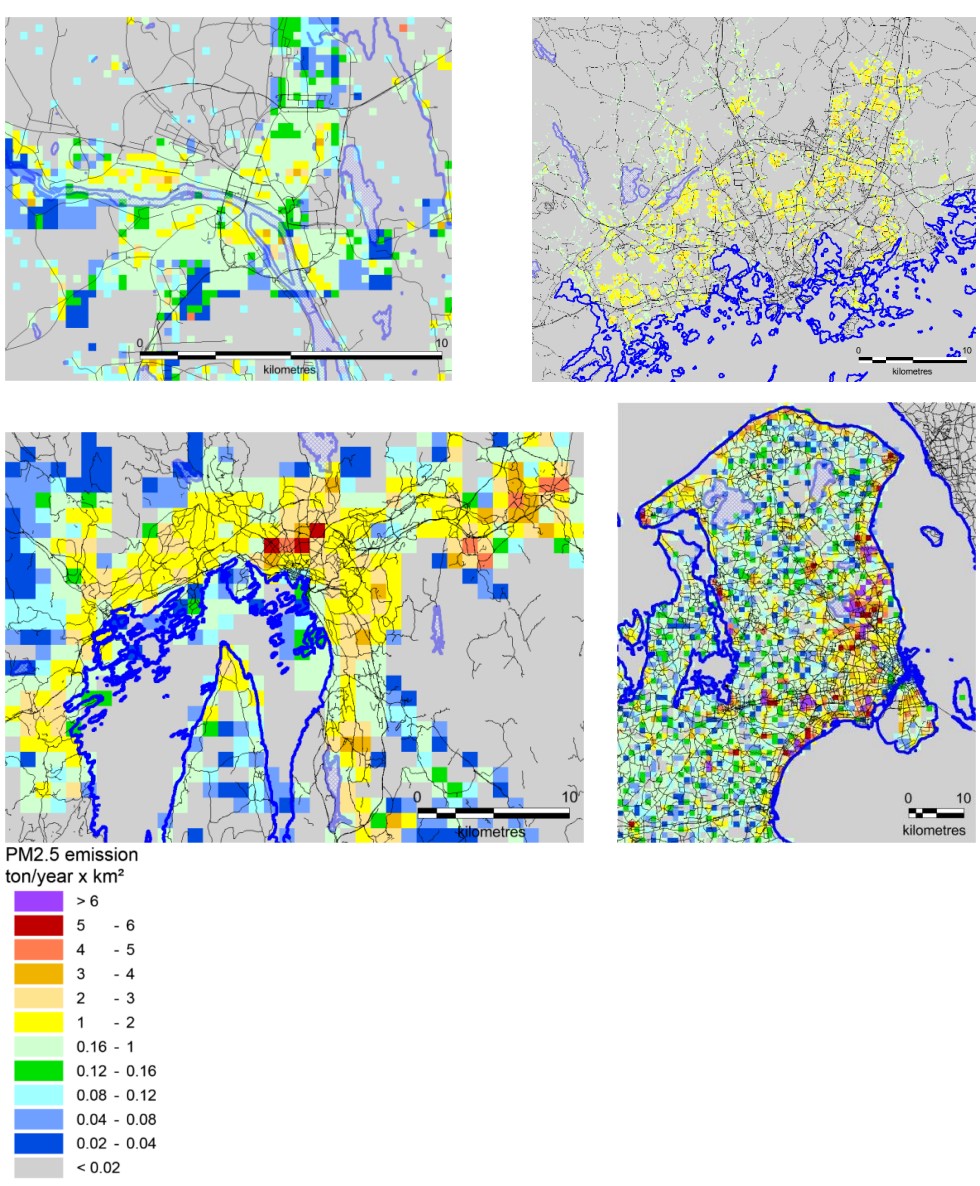


Figs. 5a-d. The predicted emissions of PM$_{2.5}$ originated from RWC in Umeå (a), Helsinki (b), Oslo (c)
and Copenhagen (d). The results represent the year 2011 for Umeå, 2013 for Helsinki and Oslo, and
2014 for Copenhagen. The spatial resolution is 250 x 250 m$^2$ for Umeå, 100 x 100 m$^2$ and Helsinki, and
1 x 1 km$^2$ for Oslo and Copenhagen. The unit is ton/(year km$^2$) for all the domains. For clarity, the sea
shore and the boundaries of the inland water areas have been shown with blue lines. The physical
scales of the domains have also been indicated.





The considered geographic domain is clearly the largest for Copenhagen, second largest for Helsinki, third largest for Oslo, and smallest for Umeå. The spatial resolution of the emissions from RWC is finer, viz. 250 m x 250 m, and 100 m x 100 m for Umeå and Helsinki, respectively, and coarser, viz. 1 km x 1 km, for both Oslo and Copenhagen.

The results show that the spatially averaged maximum emission values are the highest for the domains of Copenhagen and Oslo; these range from negligible to more than 5.0 or 6.0 ton/(year km$^2$) in some limited areas in Oslo and Copenhagen, respectively. The emission values within the domains of Helsinki and Umeå reach up to a few ton/(year km$^2$).

The highest emission values of RWC were mainly located outside the city centres of Helsinki and Copenhagen. In particular, in the Helsinki region, the highest emissions were detected in detached and semi-detached house areas; these were situated to the west, east and north of the centre of Helsinki. The detailed locations of these areas were reported by Hellen et al. (2017). For Copenhagen, the highest emission strengths were also slightly outside the most densely built city centre; highest concentrations were observed in the suburban areas of Copenhagen.

In the Helsinki area, the buildings are mainly kept warm using an extensive district heating system, and by electricity heating and by using geothermal heat pumps. However, these systems have only a minor impact on the local air quality. The district heating is mainly produced in energy plants burning fossil fuels; most of these plants have very high stacks. On the other hand, wood combustion is mainly used as a secondary heating system in detached or semi-detached houses. In addition, it is common to use fireplaces and sauna stoves in suburban detached houses. Wood combustion appliances were used in approximately 90 % of the detached houses in the Helsinki area in 2013. Helsinki was the only target city, in which sauna stoves were an important source of PM$_{2.5}$ emissions. There is a high correlation of the spatial density of the detached or semi-detached houses, and that of the emissions from RWC in the Helsinki region.

Domestic heating in the Copenhagen area is dominated by district heating, which was used in 80 % of the residences on a national level at the time. Wood combustion was most commonly used as a secondary heating method in wood stoves in residential detached or semi-detached houses as in Helsinki. Such detached houses are mainly located in suburban regions, outside the city centre. RWC in the Copenhagen area is dominated by the use of wood stoves, compared to boilers.

In addition to suburban regions, there is a significant number of wood stoves used in apartments in the city centre of Copenhagen. The stoves in these apartments have on average a lower rate of wood consumption compared to the ones in detached and semi-detached houses. Wood stoves can also be located in the cottages in allotments. The emission gridding methodology used in this study has taken into account both the differences of the rate of consumption for the different building types, and those for the RWC used as primary and secondary heating.

For Oslo and Umeå, the highest emission values from RWC were located within the city centres. Concerning Oslo, the highest PM$_{2.5}$ emissions were attributed to residential areas, which contain aged blocks of flats and multifamily dwellings, both located in the Oslo city centre and its surroundings. A major fraction of these buildings was constructed in the beginning of the 20[th] century, and wood stoves





are commonly used for heating. There were also relatively high emission densities in the densely inhabited eastern parts of Oslo and in the neighbouring municipalities to the east of Oslo.

In Umeå, the largest emissions were originated from relatively old buildings, which were not connected to the district heating system. In such buildings, wood boilers are commonly used as a primary heating source. Umeå was the only target city, in which boilers were widely used inside the urban area. Although there was a smaller number of this kind of buildings within the centre of the city, a majority of them were detached or semi-detached houses located outside of the city of Umeå. In residential areas connected to the district heating system, stoves are commonly used as a secondary heating source. The wood consumption for such stoves is considerably lower than that for boilers. However, the number of stoves is substantially higher; the stoves were therefore the main source of RWC emissions within the most recently constructed residential areas.

### 3.3 PM$_{2.5}$ concentrations and source contributions from RWC

The urban scale concentration distributions are determined by the corresponding spatial and temporal distributions of the urban emissions and the meteorological conditions. The contributions from RWC are strongly influenced by the extent of district heating systems, the distributions of residential areas, and the spatial density and types of usage of the combustion devices. The concentration distributions from RWC can be correlated with the corresponding ones of residential areas. The residential areas are often situated mainly in suburban regions; however, there can be also a substantial number of residents in the city centres or in regional urban centres.

Clearly, the dilution of pollution is dependent on the meteorological conditions during any selected year. In addition, the amount of wood combustion is influenced by the evolution of the ambient temperatures, especially during the colder winter periods. The strengths of other urban pollution sources and of the regional background are essential factors in terms of the source contributions to RWC.

The predicted PM$_{2.5}$ concentrations in ambient air have been presented in Figs. 6a-d. These include the contributions originated from all the main source categories in the four Nordic cities (Umeå, Helsinki, Oslo and Copenhagen), and the regional background concentrations. The results have been computed on fine urban scale resolutions for Umeå, Helsinki and Oslo; of the order of from 20 to 50 m in the vicinity of the local sources, such as vehicular traffic, industrial sources and energy production. In case of RWC, the spatial resolutions of the dispersion modelling were approximately the same as for the emission inventories in each city, corresponding to that source category. For the Copenhagen domain, the spatial resolution of the dispersion modelling was 1 x 1 km$^2$.





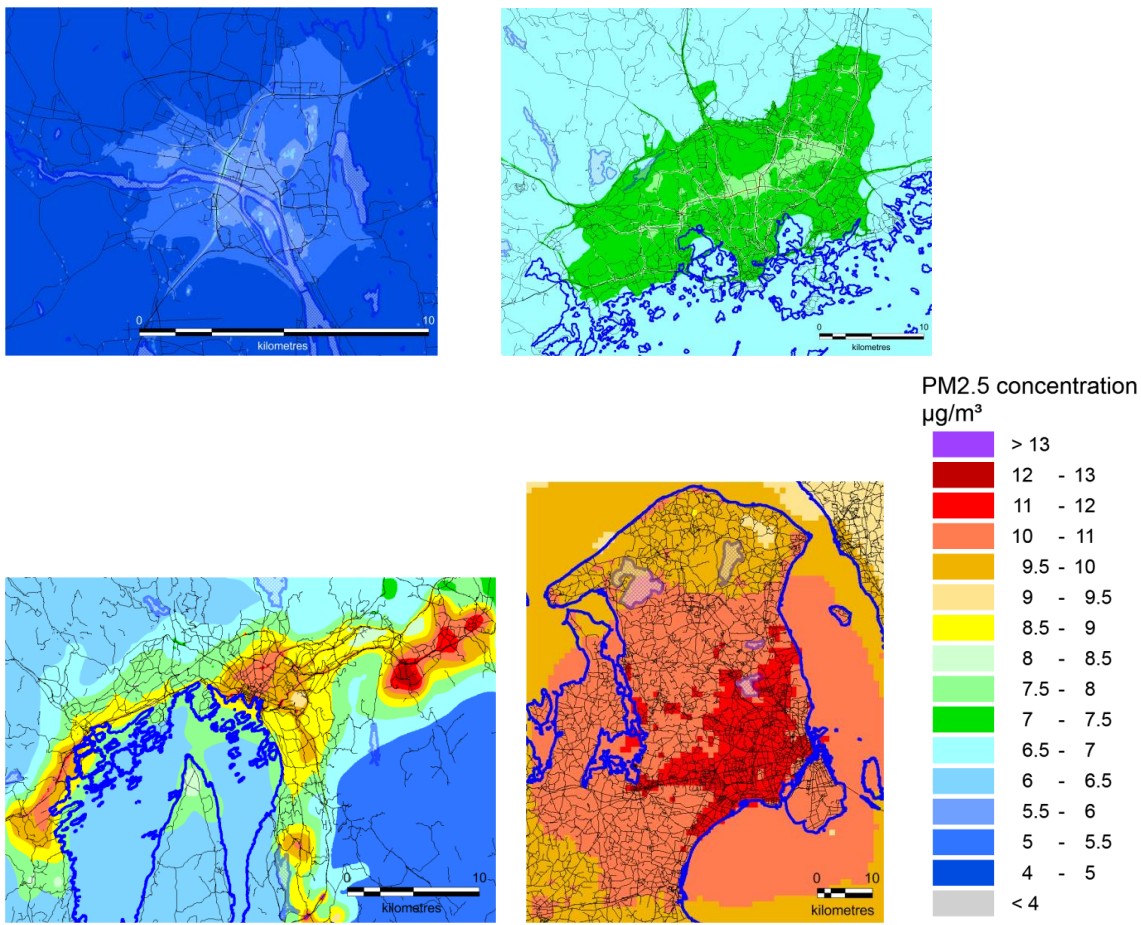

Figs. 6a-d. The predicted concentrations of PM$_{2.5}$ in Umeå (a), Helsinki (b), Oslo (c) and Copenhagen
(d). The results represent the year 2011 for Umeå, 2013 for Helsinki and Oslo, and 2014 for
Copenhagen. The main road and street networks have been presented as black lines. The sea and inland
water boundaries have been presented as blue and violet solid lines, respectively.

The concentration distributions are influenced also by the atmospheric dispersion conditions in the
cities. In particular, Oslo is surrounded by higher ground and numerous hills, which tends to reduce the
dilution of pollution, whereas the other target cities are located in a fairly flat terrain. Copenhagen and
Oslo are located in a maritime climate, whereas Umeå and Helsinki are to a larger extent influenced
also by continental climate conditions.





The annual average concentrations ranged spatially from 4 to 7 µg/m$^3$, from 6 to 10 µg/m$^3$, from 4 to more than 13 µg/m$^3$ and from 9 to more than 13 µg/m$^3$ in Umeå, Helsinki, Oslo and Copenhagen, respectively. The regional scale PM$_{2.5}$ concentrations in Denmark were higher than those in the other Nordic countries, due to the higher long-range transported contribution. Both regional background and local contributions were the lowest for Umeå. The reasons were that Umeå is clearly the smallest of the target cities as well as it is situated at larger distance from the main pollution source areas in Central, Central Eastern and Eastern Europe.

For Umeå, Oslo and Copenhagen, the highest concentrations occurred mainly in the city centres. For Helsinki, the detailed numerical data showed that the highest concentrations occurred (i) in the residential areas that are mainly situated north of the city centre and (ii) in the vicinity of the densely trafficked roads, and near the junctions of such roads. The influence of major traffic networks for all the cities are evident in the figures. Particularly. for Oslo, the overall distribution of concentrations is very similar to that of the residential areas, and the concentrations tend to be relatively higher in the areas characterized by residential houses.

The fractions of RWC contributions to the PM$_{2.5}$ concentrations of PM$_{2.5}$ within the selected domains have been presented in Figs. 7a-d. The predicted fractions originated from RWC of the concentrations ranged spatially from 0 to 15 %, from 0 to 20 %, from 8 to 30 % and from 0 to 60 % in Helsinki, Copenhagen, Umeå and Oslo, respectively. The contributions of RWC in Oslo were clearly the highest within the target cities.



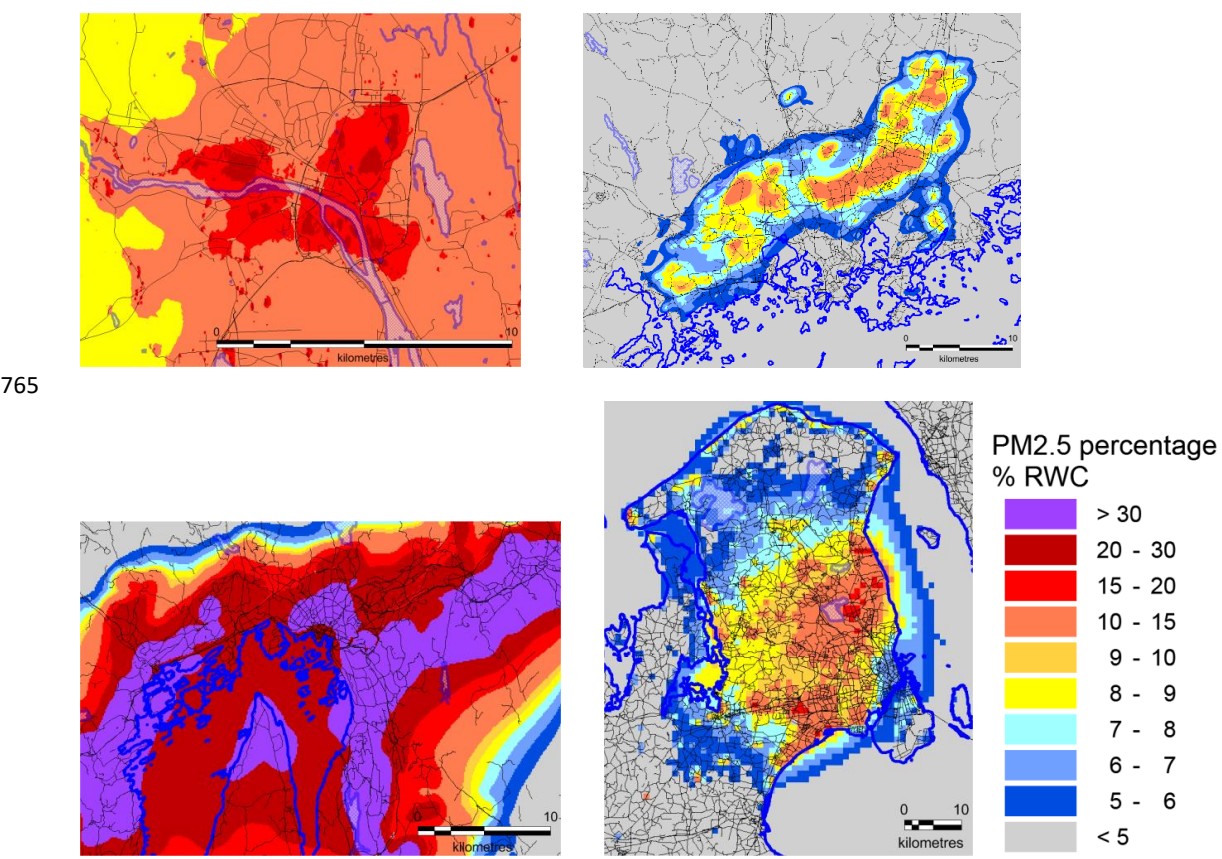

Figs. 7a-d. The spatial distributions of the source contributions of RWC to the concentrations of PM$_{2.5}$
as percentages in Umeå (a), Helsinki (b), Oslo (c) and Copenhagen (d). The results represent the year
2011 for Umeå, 2013 for Helsinki and Oslo, and 2014 for Copenhagen.

These spatial ranges of these annual average fractions have also summarized in Fig. 8., presented both
as percentages and as absolute concentration values. The RWC contributions were clearly highest for
Oslo, both in terms of absolute concentrations and their proportions of the total PM$_{2.5}$ concentrations.



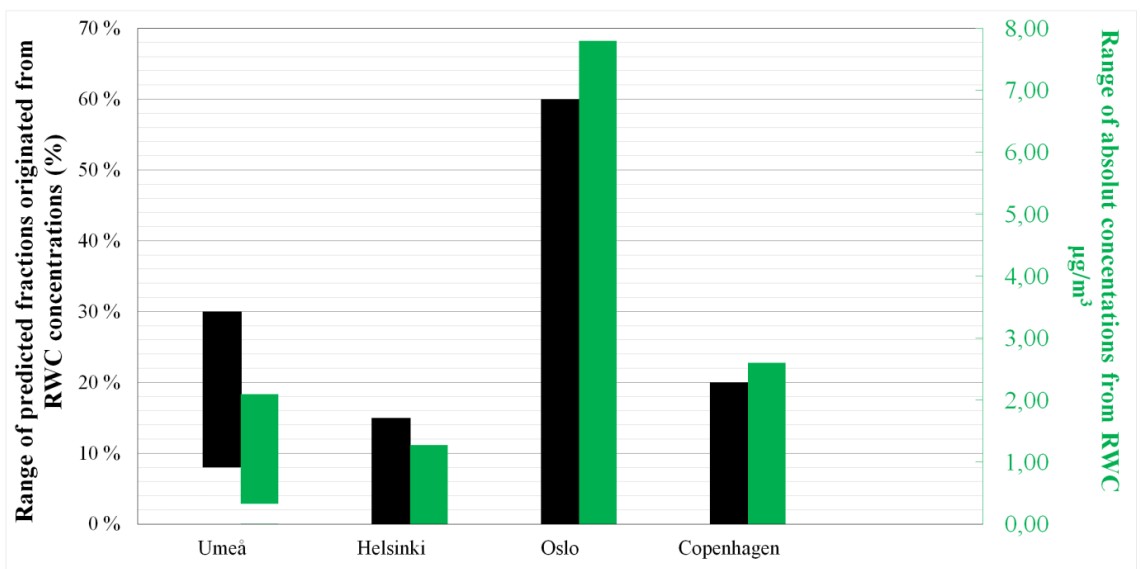

Fig. 8. The ranges of the source contributions of RWC to the concentrations of PM$_{2.5}$ as percentages (left-hand side axis) and as absolute concentrations (right-hand side axis) within the considered domains in Umeå, Helsinki, Oslo and Copenhagen. The results represent the year 2011 for Umeå, 2013 for Helsinki and Oslo, and 2014 for Copenhagen.

The fractions have been evaluated on an annual average level. In general, wood combustion is mainly used during the colder half-year, and especially during the winter months; the fractions evaluated solely for the colder periods are therefore substantially higher.

In Umeå, the highest fractions occurred both in the city centre and in the vicinity of the nearby villages with a relatively larger density of residences. In Helsinki, the highest fractions occurred in the residential areas that are mainly situated west, north and north-east of the city centre. The use of wood combustion in the centre of Helsinki is negligible. In Copenhagen, the highest fractions were located in the residential areas situated to the north and west of the city centre, similarly to the case of Helsinki.

In Oslo, the contributions of RWC ranged from negligible at the outskirts of the domain to up to 60 % in some of the easternmost parts of the domain. Wood combustion contributed more than 40 % in central Oslo. The areas, in which the annual mean PM$_{2.5}$ concentrations were the highest (Fig. 6c) coincided with the areas, in which the source contribution from RWC was the highest (Fig 7c).

The high percentages for Oslo were caused mainly by two reasons. First, wood combustion is used extensively within a fairly limited area (compared to the other target cities), for both the heating of smaller detached or semi-detached houses and for larger blocks of flats. The shares of wood combustion in Oslo were recently studied based on citizen involvement (Lopez-Aparicio, 2017c). It was found that 46% of the wood used for residential heating was applied in residences in blocks of





flats. Such flats use commonly wood stoves or open fireplaces as heating installations. The rest of the
        wood combustion took place in detached and semidetached houses, duplexes and townhouses.

        The fractions of the PM$_{2.5}$ concentrations originated from RWC in Oslo were similar to those that have
        been previously evaluated by Tarrason et al. (2018) for other Norwegian cities. Tarrason et al. (2018)
        also evaluated such fractions using emission inventories, combined with dispersion modelling.
According to that study, the annually averaged fractions from RWC in the areas of 13 Norwegian cities
        ranged from 20 to 75 %.

        In principle, one or more of the considered years could be meteorologically exceptional or very rare, in
        terms of the ambient air temperatures. This could have a substantial effect on the use of wood
        combusted. We have therefore analyzed the seasonal variation of temperatures in the selected cities
during four years. The main results of this analysis are presented in Appendix C. It can be concluded
        that none of the considered years was exceptional or rare in any of these cities, in terms of the seasonal
        variation of the ambient temperatures.

## 4. Conclusions

We have evaluated the emissions and ambient air concentrations of fine particulate matter in four
        Nordic cities, with a special emphasis on the contributions originated from small-scale RWC. The
        influence of RWC has not previously been analyzed in a harmonised manner for several Nordic cities.

        The reliable and accurate assessment of the emissions from RWC still remains a challenge. Wood
        combustion emissions are commonly known less accurately than those from most other source
categories, such as, e.g., vehicular traffic, larger scale energy production or industry (e.g., Karvosenoja,
        2008). Uncertainties are caused, amongst other factors, by the spatial distributions of pollution sources,
        activity data of wood combustion, and the emission factors.

        The numerical predictions were evaluated against measured urban scale data regarding the PM$_{2.5}$
        concentrations in the four target cities. The fractional bias values were reasonably good, within the
range from -0.20 to + 0.20, for all the regional and urban background values, and for most of the traffic
        and RWC stations. The agreement of the daily modelled and measured time series was also fairly good,
        > 0.55, in most cases. For Helsinki and Oslo, finer resolution computations were conducted and the
        data of several RWC and vehicular traffic influenced stations were available. However, there was no
        substantial systematic difference between the model performances at vehicular traffic stations,
compared with those at the corresponding RWC stations.

        The spatially averaged maximum emission values were the highest for Copenhagen and Oslo; these
        range from negligible to more than 5.0 or 6.0 ton/(year km$^2$) in some limited areas in Oslo and
        Copenhagen, respectively. The emission values within the domains of Helsinki and Umeå reach up to a
        few ton/(year km$^2$). The highest emissions from RWC were mostly located outside the city centres for
Helsinki and Copenhagen. In the Helsinki region, the highest RWC emissions occurred in detached and
        semi-detached house areas, which were to the west, northeast and north from the centre of Helsinki.
        For Copenhagen, the highest emission strengths were found also outside the most densely built city





centre. In the Helsinki area, there is an extensive district heating system, and wood combustion is mainly used as a secondary heating system in detached or semi-detached houses. The use of wood combustion in the centre of Helsinki is negligible. This explains the strong correlation between the spatial density of the detached or semi-detached houses and emissions from RWC.

With regard to Umeå and Oslo, the highest emission values from RWC were located within the city centres. In particular, in Oslo, the highest $PM_{2.5}$ emissions correspond to residential areas, which include aged blocks of flats and multifamily dwellings. There are also relatively high emission densities in the densely inhabited eastern parts of Oslo and in the neighbouring municipalities situated east of Oslo. In Umeå, the largest emissions were originated from relatively older buildings, which have not been connected to the district heating system. In such buildings, wood boilers were commonly used as a primary heating source. Most of these buildings were detached or semi-detached houses located outside the city of Umeå, although there was a smaller number of them within the centre of the city.

Both the measured and predicted concentration values were the highest for Copenhagen, caused mainly by the relatively higher regional background contributions, compared with the other three cities. The concentrations are second highest for Oslo, mainly due to substantial urban contributions. The annual average $PM_{2.5}$ concentrations ranged spatially from 4 to 7 µg/m$^3$, from 6 to 10 µg/m$^3$, from 4 to more than 13 µg/m$^3$ and from 9 to more than 13 µg/m$^3$ in Umeå, Helsinki, Oslo and Copenhagen, respectively.

For Umeå, Oslo and Copenhagen, the highest concentrations occurred mainly in the city centres. However, for Helsinki, the highest concentrations occurred in the suburban residential areas and in the vicinity of the densely trafficked roads. Major traffic networks had a substantial influence on the air quality for all these four cities. For Oslo, the spatial distribution of concentrations was very similar to that of the residential areas.

The annual average fractions of RWC contributions to the concentrations of $PM_{2.5}$ ranged spatially from 0 to 15 %, from 0 to 20 %, from 8 to 30 % and from 0 to 60 % in Helsinki, Copenhagen, Umeå and Oslo, respectively. The contributions of RWC in Oslo were clearly the highest compared to the target cities. In Umeå, the highest fractions occurred both in the city centre and in the vicinity of the nearby villages with a relatively larger density of residences, whereas in Helsinki, the highest fractions occurred in the suburban residential areas. In Oslo, the RWC contributions were up to 60 % in some of the easternmost parts of the domain, and larger than 40 % in central Oslo. In Copenhagen, the highest fractions were also located slightly outside the city centre, similarly to the case of Helsinki.

The high percentages of the contributions of RWC in Oslo were mainly attributed to the fact that wood combustion is used extensively within a fairly limited area (in comparison to the other target cities), and it is used both for the heating of smaller detached or semi-detached houses, and for larger blocks of flats.

Whereas attempts have been made to regulate RWC in the Nordic countries, there are grounds for increased policy and technical measures, to avoid and alleviate harmful impacts of RWC, including especially those on human health. Regulation should consider the whole chain of events from emissions through population exposures to impacts. Both generic and more specific, targeted measures



will be useful, in connection with general policies on air pollution, the environment, energy and climate, and those on urban planning, housing and buildings. The range of measures could include regulatory ones, information campaigns and economic steering, and their combinations.

# 5. Code and data availability

The SILAM code is publicly available.

The emission data, and the measured and predicted concentration data used in this study is available, by contacting the responsible authors in each country, i.e., J. Kukkonen, C. Geels, S. López-Aparicio and D. Segersson.

# 6. Acknowledgements

This study has been part of the project "Understanding the link between Air pollution and Distribution of related Health Impacts and Welfare in the Nordic countries", project #75007 (NordicWelfAir), funded by Nordforsk. We also acknowledge the funding of the Academy of Finland for the projects "Global health risks related to atmospheric composition and weather" (GLORIA) and "Environmental impact assessment of airborne particulate matter: the effects of abatement and management strategies" (BATMAN). This study was also part of the Swedish Clean Air and Climate research programme (SCAC), which has been funded by the Swedish Environmental Protection Agency.

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





**Appendix A. The assessment of emissions from other source categories in the target cities, in addition to RWC**

**A1. Umeå**

Exhaust emissions originated from vehicular traffic were estimated using measured traffic flow information, the traffic flow model EMME/2, and the emission factors of Handbook Emission Factors
for Road Transport by Hausberger (2009). Measured traffic flow information included separately light and heavy duty vehicles; this information was complemented with predictions provided by a traffic flow model. The information on the vehicle fleet composition was derived based on the national vehicle registry; however, this information was refined to allow for the local information regarding the share of heavy vehicles.

A resuspension model by Omstedt (2005) was applied to evaluate the non-exhaust emissions. The model can be used to analyze also the wear due to studded tyres and street sanding.

Emissions from the category of other sources were extracted from the yearly national compilation of spatially distributed emissions in Sweden. The emissions originated from shipping were evaluated using the SHIPAIR model (Segersson, 2014). The largest contributions from the other sources were
originated from off-road machinery and major point sources (Segersson et al., 2017).

**A2. Helsinki**

An emission inventory for vehicular traffic for 2013 was used. This inventory included both traffic exhaust and traffic suspension emissions for the network of roads and streets in the Helsinki Metropolitan Area (HMA). The spatial distribution of vehicular emissions was based on detailed
information on the line source network in the HMA provided by the Helsinki Region Transport. The number of line sources in the revised inventory was 26 536. The traffic volumes and average travel speeds at each traffic link were computed using the EMME/2 transportation planning system for three time periods of the day (HSL, 2011). The hourly traffic volumes were computed using a set of regression-based factors.

The total $PM_{2.5}$ exhaust emission values in the HMA for 2013 were estimated using data of the national calculation system for traffic exhaust emissions and energy consumption in Finland, called LIPASTO (Mäkelä and Auvinen, 2009), containing city-level data on emissions and mileages for various classes of vehicle types, and for streets and roads.

We have evaluated the hourly vehicular suspension emissions of $PM_{2.5}$ using emission factors
computed using the FORE model (Kauhaniemi et al., 2011, 2014), which is based on the resuspension model by Omstedt (2005). The same traffic mileage data was applied as for the estimation of exhaust emissions.

**A3. Oslo**

We have considered the most important local emission source categories, viz. RWC, on-road and non-
road traffic, industry and shipping (López-Aparicio et al., 2017a). The emissions have been evaluated for the year 2013.





On-road traffic emissions were estimated at the road links, taking into account road type, width, length, the average daily traffic, the road vehicle distribution as vehicle class and vehicle technology class. The baseline emission factors were selected based on the Handbook Emission Factors for Road Transport (HBEFA, 2010), and they are adjusted based on the ageing of the vehicle, as a function of the mileage, and factors that relate to speed dependency.

Vehicular non-exhaust emissions of $PM_{2.5}$, due to suspension of road dust, were calculated based on a simplified version of the NORTRIP model (Denby et al., 2013). The NORTRIP model can be used to compute road surface moisture and dust production, dust loadings and suspended particulate emissions to the air. In its original form, the NORTRIP model is used to calculate the surface moisture separately for each and every road in the considered domain. However, when the model was applied in a simplified way, it was used to compute the surface moisture for two road categories: (i) a characteristic heavily trafficked road with salting and (ii) a less densely trafficked road without salting. The moisture at every road is then evaluated as a weighted average, depending on the road type.

Emissions from shipping were estimated based on the detailed shipping activity data from the port of Oslo following a bottom-up approach (López-Aparicio et al., 2017b). The emissions were computed following the method suggested by US EPA (2009). These are based on detailed information regarding the individual vessels visiting the port, the emission factors for each vessel category and operational modes of ships. The modelled industrial emissions consisted of emissions from point sources and diffuse emissions. The emissions from off-road mobile combustion included construction machinery, tractors, households and gardening.

### A4. Copenhagen

The assessment of other emissions for the Danish area were based on the SPREAD model (Plejdrup et al., 2016; Plejdrup and Gyldenkærne, 2011). The main emission sectors included were stationary combustion, mobile combustion, fugitive emissions from fuels, industrial processes and product use, agriculture and waste. The SPREAD model evaluates yearly average emissions.

In this study, the road transport emissions were used as included in the national emission inventory within the SPREAD model. The emission factors were based on the COPERT V model, which was adapted to national conditions (Nielsen et al., 2017). The spatial distribution of the national road transport emissions was based on the Danish national GIS-based road network and traffic database, which includes mileage data in terms of road type and vehicle composition.





## Appendix B. A more detailed description of the comparison of model predictions and measured data.

### B1. Evaluation of the predicted concentrations for Umeå

For the model evaluation, we used the concentration data from a regional background station (Bredkälen), an urban background station (Biblioteket), one station in vehicular traffic environment (Västra Esplanaden), and four stations specially located to measure the contributions from RWC (Sävar, Vännasby, Vännas and Tavleliden). The results of these comparisons are presented in Table B1. The model computations have been performed for the years 2006-2013 (Omstedt et al., 2014, Segersson et al., 2017).

Table B1. Selected statistical parameters on the agreement of predictions and measurements for the daily concentrations of PM$_{2.5}$ in the Umeå area. The measured values at the station of Bredkälen have been adjusted slightly, based on regional-scale dispersion model computations. For Bredkälen, the results for two different periods have been separately presented.

| Name of the station | Classification | Observed mean ($\mu g/m^3$) | Predicted or adjusted (in case of Bredkälen) mean ($\mu g/m^3$) | Fractional bias | Number of data points | Measurement period |
|---|---|---|---|---|---|---|
| Bredkälen | Regional background | 1.93 | 3.97 | - | 1389 | 2009 -2012 |
| Bredkälen | Regional background | 1.90 | 3.18 | - | 196 | Nov 2012 – May 2013 |
| Biblioteket | Urban background | 4.90 | 5.70 | 0.16 | 49553 | 2006-2012 |
| Västra Esplanaden | Traffic | 7.80 | 11.60 | 0.39 | 48211 | 2006-2011 |
| Sävar | RWC | 3.60 | 4.30 | 0.18 | 399 | Nov 2012 - Dec 2013 |
| Vännasby | RWC | 4.20 | 4.80 | 0.13 | 286 | Nov 2012 - Dec 2013 |
| Vännas | RWC | 6.10 | 4.50 | -0.30 | 25 | Nov 2012 - May 2013 |
| Tavleliden | RWC | 3.80 | 3.50 | -0.08 | 20 | Jan 2013 - May 2013 |

The values include the annual average concentrations for the years 2006-2011 at the stations in the city of Umeå (Västra Esplanaden and Biblioteket). For the other stations (Sävar, Vännasby, Vännas and Tavleliden), the measured values are based on daily or weekly samples during the above mentioned periods. The measurement sites represent a densely trafficked street canyon (Västra Esplanaden), urban background (Biblioteket), and residential environments (Sävar, Vännasby, Vännas and Tavleliden).

The agreement of the measured and modelled long-term average values can be considered to be fairly good for all the sites, except for the street canyon site (Västra Esplanaden).





The method used for the evaluation of the temporal variation of concentrations originated from RWC uses the measured concentration values of PM$_{2.5}$. We therefore cannot independently evaluate the performance of the model, with regard to the temporal correlations of the measured and predicted time series of concentrations.


### B2. Evaluation of the predicted concentrations for Helsinki

For the model evaluation, we used the concentration data from the following stations: regional background station of Luukki, the urban background station of Kallio, three stations in vehicular traffic environments (Mannerheimintie, Leppävaara, Tikkurila), and three stations specially located to measure the contributions from RWC (Vartiokylä, Tapanila and Kauniainen). The results of these comparisons are presented in Table B2.

Table B2. Selected statistical parameters on the agreement of predictions and measurements for the daily concentrations of PM$_{2.5}$ in the Helsinki area in 2013. Notation: RWC = RWC.

| Name of the station | Classification | Observed annual mean ($\mu g/m^3$) | Predicted annual mean ($\mu g/m^3$) | Index of agreement | Factor-of-two (%) | Fractional bias | Number of data points |
|---|---|---|---|---|---|---|---|
| Luukki | Regional background | 6.3 | 6.7 | 0.59 | 58 | 0.05 | 364 |
| Kallio | Urban background | 7.0 | 7.2 | 0.64 | 65 | 0.04 | 364 |
| Mannerheimin-tie | Traffic | 8.6 | 7.6 | 0.60 | 64 | -0.13 | 363 |
| Leppävaara | Traffic | 7.1 | 8.1 | 0.68 | 74 | 0.12 | 363 |
| Tikkurila | Traffic | 7.2 | 7.8 | 0.73 | 75 | 0.08 | 363 |
| Vartiokylä | RWC | 6.8 | 7.5 | 0.69 | 68 | 0.10 | 351 |
| Tapanila | RWC | 9.1 | 7.8 | 0.67 | 65 | -0.16 | 360 |
| Kauniainen | RWC | 7.1 | 7.3 | 0.65 | 65 | 0.03 | 360 |


Overall, the modelled PM$_{2.5}$ concentrations agreed fairly well or well with the measured data. The values of the index of agreement (IA) and the factor-of-two (F2) were slightly lower at the regional background station of Luukki, compared with the corresponding values for the urban stations.

The range of model performance was similar at the three traffic stations compared to the corresponding performance at the RWC–influenced stations. For instance, the IA values ranged from 0.60 to 0.73, and from 0.65 to 0.69 at the traffic and RWC –influenced stations, respectively. Concerning the traffic station of Mannerheimintie in the center of the city, there is an under-prediction (FB = - 0.13), which can be attributed to the reduced dilution caused by buildings and the frequent congestion of traffic. There is also under-prediction at the station of Tapanila (FB = – 0.16), located in a residential area.






**B3. Evaluation of the predicted concentrations for Oslo**

The modelled regional background $PM_{2.5}$ concentrations were compared to the regional background $PM_{2.5}$ measurements at the station of Hurdal in southern Norway. The mean fractional bias varied from

-0.54 to 0.65 during the whole year. We noticed that the modelled ensemble results in $PM_{2.5}$ weekly means were lower in summer when compared to the measurements, whereas during the rest of the seasons they are remarkably higher, especially from October to December. These differences might be explained by (i) the inaccuracies related to partially missing secondary organic aerosol formation in the model ensemble, (ii) the inaccuracies in modelling particulate matter originated from biogenic sources

(Aas et al., 2014), and (iii) uncertainties in primary aerosol emissions (Marécal et al, 2015). The predicted regional background concentrations were based on an ensemble of seven chemical transport models, four of which did not include secondary organic aerosol formation processes. The contribution of the background concentrations to the $PM_{2.5}$ results within the city of Oslo is on average 56%.

For the urban scale model evaluation we utilized the concentration data from all the available

permanent measurement stations within the selected domain in 2013. All the stations in the area were designed as traffic stations, except for one urban background station. Even though there are no stations which could be used to measure the contributions of RWC, Akerbergveien, Bygdoy Alle and Kirkeveien traffic monitoring stations could be considered as the most influenced by RWC emissions. These stations are located in urban roads surrounded by residential areas (i.e., blocks of flats)

characterized by intense wood burning activity. The selected statistical parameters of this evaluation are presented in Table B3.

Table B3. Selected statistical parameters on the agreement of predictions and measurements for the daily concentrations of $PM_{2.5}$ in the Oslo area in 2013.

| Name of air quality station | Classification | Observed annual mean ($\mu g.m^{-3}$) | Modelled annual mean ($\mu g.m^{-3}$) | Index of agree-ment | Factor-of-two (%) | Fractional bias | Number of data points |
|---|---|---|---|---|---|---|---|
| Hurdal | Regional | 3.1 | 3.6 | 0.33 | 59 | 0.12 | 51 |
| Sofienbergparken | Urban background | 11 | 10 | 0.72 | 67 | -0.096 | 365 |
| Akebergveien | Traffic | 9.2 | 9.9 | 0.82 | 83 | 0.072 | 335 |
| Alnabru | Traffic | 16 | 10 | 0.63 | 66 | -0.43 | 356 |
| Bygdoy Alle | Traffic | 12 | 9 | 0.74 | 71 | -0.33 | 365 |
| Hjortnes | Traffic | 9.4 | 9.4 | 0.79 | 89 | 0.0028 | 364 |
| Kirkeveien | Traffic | 8.6 | 8.0 | 0.73 | 83 | -0.062 | 345 |
| Manglerud | Traffic | 9.0 | 10 | 0.71 | 88 | 0.15 | 361 |
| RV4 Aker Sykehus* | Traffic | 9.1 | 10 | 0.65 | 78 | 0.14 | 201 |
| Smestad* | Traffic | 10 | 7.9 | 0.83 | 92 | -0.27 | 193 |

* RV4 Aker Sykehus and Smestad were not in operation from May to mid-October.





The model simulation results have been benchmarked with the DELTA tool (http://aqm.jrc.ec.europa.eu/index.aspx), which has been developed by the Joint Research Centre in the framework of the FAIRMODE action (Forum for air quality modelling in Europe; http://fairmode.jrc.ec.europa.eu/). The analysis that the results of all considered air quality stations in Oslo fulfill the Model Quality Objectives, as defined by this assessment tool.

Results showed that the comparison of the model performance and the agreement between measurements and predictions amongst traffic stations was poorer for the station of Alnabru. This station is located between two roads in a valley, along where winds from the Oslo fjord frequently transport substantial pollution from central Oslo.

As expected, the highest $PM_{2.5}$ concentrations were observed in winter and daily values at the urban background station (Sofienbergparken) can be higher than 40 $\mu gm^{-3}$. According to a survey on wood combustion by Statistics Norway, emissions from RWC are diurnally the highest in the evening.

**B4. Evaluation of the predicted concentrations for Copenhagen**

For the model evaluation, we exploited the concentration data from the regional background station of Risø, the urban background station of HCØ (H.C. Ørsted Institute), two stations in vehicular traffic environments (JGTV, Jagtvej and HCAB, H.C. Andersens Boulevard), and one station in a suburban area (Hvidovre). The latter site was selected to represent the influence of residential small-scale combustion. The results of these comparisons are presented in Table B4.

Table B4. Selected statistical parameters on the agreement of predictions and measurements for the daily concentrations of $PM_{2.5}$ in the Copenhagen area. The results correspond to the period 2013-2017, except for the stations of JGTV and Hvidovre, for which the measurements were started later on, in November, 2013 and in June, 2015, respectively. Notation: RWC = RWC.

| Name of the station | Classification | Observed annual mean ($\mu g/m^3$) | Predicted annual mean ($\mu g/m^3$) | Index of agreement | Factor-of-two (%) | Fractional bias | Number of data points |
|---|---|---|---|---|---|---|---|
| Risø | Regional background | 10.5 | 10.0 | 0.70 | 95 | -0.05 | 1753 |
| HCØ | Urban background | 11.6 | 9.7 | 0.68 | 93 | -0.18 | 1649 |
| JGTV | Traffic | 14.7 | 9.7 | 0.59 | 82 | -0.41 | 1415 |
| HCAB | Traffic | 16.6 | 9.8 | 0.45 | 70 | -0.52 | 1638 |
| Hvidovre | RWC | 10.3 | 9.5 | 0.68 | 94 | -0.09 | 870 |

Regarding regional background concentrations, the model predicted well both the long-term averages (FB = - 0.05) and their daily variability (IA = 0.70). For the urban background site, the model predictions were also good or fairly good (FB = - 0.18, IA = 0.68).

The modelling was done using a spatial resolution of 1 x 1 $km^2$. The concentrations at the two vehicular traffic sites were substantially underestimated (FB = - 0.41 and – 0.52). The agreement of the



daily temporal variations at the traffic sites were also lower than the corresponding one for the other selected stations. With respect to the site affected by RWC, the model performance was fairly good.



**Appendix C. Evaluation of the variations of the ambient temperatures during the period 2011 –**
**2014 in the four selected cities.**

The seasonal variations of temperatures at four measurement stations in the four cities have been presented in Figs. A1 a-d. The data has been extracted from the open data portals of the Swedish, Finnish, Norwegian and Danish meteorological institutes. The selected stations can be considered to be representative for the meteorological conditions in these cities. For Copenhagen, the earliest part of the
data, i.e., from 1 Jan, 2011 to 30 May, 2012 has been interpolated, based on data from several meteorological stations in Denmark.

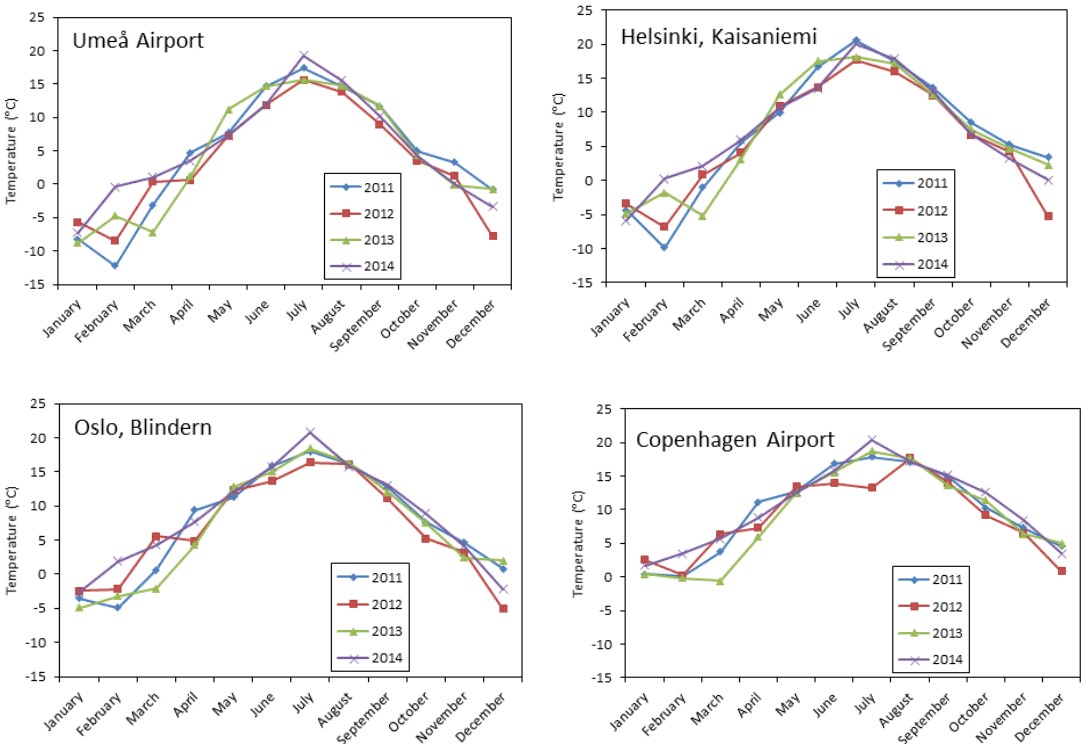

Figs. C1 a-d. The monthly averaged values of the measured ambient temperatures in the four selected cities, for the years 2011 – 2014.

None of the considered years can be considered to be exceptional or rare at any of these locations, in terms of the ambient temperatures.




**Appendix D. An overview of the regulatory frameworks for RWC in four Nordic countries**

We have addressed here both the policies issued by the EU and those by the four considered Nordic countries. Sweden, Denmark and Finland are EU member states, whereas Norway is an associated country to the EU. In the EU member-states and associated countries, EU directives are nationally translated into laws and national regulations.

Whereas air quality policies at a general level are similar across the Nordic countries, the national governance systems and political cultures differ, and thus the EU regulation is translated to national policy and regulation in different ways. Policy and regulation that manage air pollution originated from RWC is linked to regulation of clean air, urban environments and the heating of houses in the Nordic countries.

In the following, first, the relevant EU regulations are briefly reviewed, second, the regulations in each country are addressed, and third, the implemented and possible future regulations and measures are discussed.

**D1. The EU regulations**

The Clean Air Act and its successors set a general framework for the assessment and improvement of air quality in the EU. However, it does not specifically target the contributions from RWC. The main directives were merged into the Clean Air Directive (2008/50/EC) on Ambient Air Quality and Cleaner Air for Europe, which aimed to establish clean air at a level, which was not considered to be harmful to human health and nature. The Clean Air Directive requires the member states to lower the level of exposure to $PM_{2.5}$ by 20 % by 2020, relative to the levels in 2010.

**D2. The RWC regulations in four Nordic countries**

**D2.1 Finland**

Finland issued its first national programme for the protection of air quality in 2002, complying with Directive 2001/81/EU. Whereas the emissions from many sources of air pollution, such as industry, large-scale energy production and most areas of transport have been regulated and reduced since that time, emissions from RWC have been substantially less regulated. As part of the implementation of Emissions Ceiling Directive by EU in 2016, a national programme for the protection of air quality has been issued in 2018.

The above mentioned national programme (2018) has assessed the compliance with national emission ceiling (NEC) directive, has proposed recommendations for further actions to reduce health impacts and environmental damage caused by air pollution, and has considered policy instruments for limiting emissions in general terms. Recommendations that addressed air emissions from RWC included the promotion of information campaigns for citizens, setting the basis for national eco-label for wood-heated sauna stoves and improving the possibilities for local authorities to intervene in RWC smoke issues.

The EcoDesign directive of boilers (EU 2015/1189) and space heating appliances (EU 2015/1185) using solid fuels will regulate most of the RWC combustion appliances used in Finland; however, stoves in saunas will be excluded. The EcoDesign regulations target energy efficiency and emissions of


new wood combustion appliances sold after 2022. However, the regulations have not extensively addressed wood burning habits nor old existing appliances. The Ministry of Health and Social Welfare has stressed the importance of emissions from wood stoves and the need to limit the associated emissions.

The authorities in some cities, such as Helsinki Metropolitan Area (HMA), have also called for
improved control of wood stoves (e.g., Kaski et al., 2016). Concrete actions include information campaigns for proper storage of fuels and use of wood stoves in HMA.

**D2.2 Denmark**

Policies and policy actions that target emissions from wood stoves emerged on Danish political agendas in the early 2000's. By 2001, the Environmental Protection Agency issued the Woodstove
Regulation, which allocated the authority to regulate the use of woodstoves in cases of severe pollution to local governments. The Woodstove Regulation in 2018 revised the regulation of use and emissions from woodstoves. Implementation of major changes in combustion plants are required prior to installation to document compliance with emission standards. The use of petro coke has not been allowed in private households since the beginning of 2019.

Policy instruments to implement the EU standards and the Danish clean air policy include also standards regarding clean burning in woodstoves. Monitoring of compliance is divided between the Environmental Protection Agency and local governments; this involves a system of monitoring stations. In monitoring of compliance among individual households, local planners are dependent on the smell reported by citizens.

Based on the Danish Planning Act, local governments have some possibilities to regulate the use of wood stoves in specified zones. The Woodstove Regulation in 2018 widens the legal options for local governments to issue plans and regulations for establishment and use of combustion plants. Local governments have over the past years called for a change of law such that local authorities could ban woodstoves in specified residential areas. In early 2018, 12 of the 98 Danish Local Governments had
specified regulations for the use of woodstoves, addressing how and what households may feed their wood stoves, including details on the operation of fuels and stoves.

RWC is gradually entering Danish policy agendas; also a public perception is emerging regarding the fact that health harming particles are not only associated with traffic. There have also been information campaigns that were based on the experiences of professional chimney sweepers. In the early 2000's,
the government has presented a subsidy scheme, in which woodstove owners have been rewarded for changing woodstoves that are older than 1990 to new and cleaner technology. This has to a large extent been effective, most of the older than 1990 wood stoves have been substituted.

**D2.3 Norway**

As an associated country to the EU, Norway follows the EU directives and regulations; whereas it
implements those independently. Ambient air quality policy therefore at least complies with The Ambient Air Quality Directive of 2016. At local level, air quality is regulated through three separate mechanisms, the Pollution Act, the national air quality objectives specified by the government and the



air quality standards. The Norwegian Government's national targets for local air quality aim at air quality that will not be harmful to human health.

The main policy instruments to implement the objectives comprise a system of monitoring with stations dispersed in Norway, and standards for emissions and behavioural instruments, including information campaigns. In several municipalities, there are economic incentives to replace older wood stoves by new installations. For instance, in Oslo municipality since 1998, there has been a payment plan for this purpose. It has been estimated that from 1998 to 2015, approximately 8677 wood stoves have been
replaced using the granted support.

The secondary homes in Norway are commonly heated by wood stoves; however, these are located in sparsely populated areas. The exposure to emissions from these wood stoves is therefore limited.

**D2.4 Sweden**

Sweden is also to a considerable degree dependent on woodstoves for heating in houses, businesses and
public institutions. The $PM_{2.5}$ levels are highest in Stockholm, which is reflected in policies and regulations; these have a specific focus on reducing emissions in cities. The Government New Strategy for Clean Air in the 2010's recognized the considerable health risks associated with $PM_{2.5}$ and identified wood stoves as a significant source of $PM_{2.5}$ emissions.

The Planning and Building Act, administered by the National Board of Housing Building and Planning,
specifies zoning and allows local governments to limit the use of RWC and to ban the wood burning practices with the highest emissions. A range of Swedish cities, towns and local governments use this regulatory tool to reduce emissions from RWC, especially in densely populated areas. For instance, the authorities in the city of Malmø have extensively applied zoning to limit the air pollution from woodstoves. The city has prohibited the so-called conveniency use of woodstoves during the warmer
months from April to September.

**D3. Discussion on RWC regulations and measures in four Nordic countries**

The contributions to air quality from all source categories in the four target countries are regulated based on the EU Clean Air Directive. The maximum allowed concentration levels therefore need to comply with the same values, and the compliance is monitored using similar measurement networks.
However, woodstoves are additionally specifically targeted by a range of national regulations in the four Nordic countries.

The possible regulatory policy instruments could include (i) national requirements for local governments to consider air quality regulations based on spatial zoning, and (ii) the ban of wood stoves in high- to medium-risk urban areas, especially residential areas. This may imply amendments to
National Planning Acts on an urban level. The economic policy instruments could include (i) subsidies for substitution to wood stoves based on cleaner burning technology at household level, and (ii) integration of the social costs, including health costs, of wood burning in the price of RWC technology, maintenance and fuels. The incentive-based measures could include (i) research and innovation to develop cleaner wood burning technology and to improve assessment of RWC exposures, in particular
targeting high-risk areas, and (ii) information campaigns to raise awareness and develop skills on cleaner wood burning practices. The campaigns could be targeted at citizens and households, sellers of



wood stoves, and chimney cleaners. Regulations have also been suggested that would ban the use of wood stoves in selected residential or urban areas during peak pollution periods. However, this is not currently allowed by the national laws in the considered Nordic countries.

Regulation of $PM_{2.5}$ emissions aims to reach urban air concentrations that are below the legally set limits, using methods, such as regulations based on zoning, cleaner technology, subsidies and information campaigns. Spatial zoning regulations, combined with a direct regulation of the use of woodstoves, have turned out to be an efficient regulatory tool in urban areas. For instance, the authorities in Sweden have used this method in urban areas, in combination with detailed specification

of when, how and how often wood stoves can be used. However, the scope of spatial zoning regulations is limited by other national and local regulations and plans.

Promotion of cleaner technology has been successfully supported in some Nordic countries through subsidies to households that voluntarily exchange older stoves to newer, technologically more advanced ones. Information campaigns have also been organized regarding the proper storage of fuels,

and the use of wood stoves.





## 8. Author contributions

Jaakko Kukkonen has coordinated the analyses, compiled together the information and written a substantial fraction of the article. David Segersson, Gunnar Omstedt, Camilla Andersson and Bertil
Forsberg have provided the relevant information and analyses on the measurements and modelling in Umeå. Camilla Geels, Ulas Im, Jesper H. Christensen, Ole-Kenneth Nielsen, Marlene S. Plejdrup, Jacob Klenø Nøjgaard and Jørgen Brandt have provided the relevant information regarding Copenhagen. Leena Kangas, Mari Kauhaniemi, Ari Karppinen, Mikhail Sofiev and Heidi Hellen have provided the relevant information regarding Helsinki. Susana Lopez-Aparicio, Gabriela Sousa Santos
and Ingrid Sundvor have provided the relevant information regarding Oslo. Kari Riikonen, Juha Nikmo and Androniki Maragkidou have worked on the processing, analysis and harmonisation of the datasets for all the  target cities. Anne Jensen, Timo Assmuth and Niko Karvosenoja have written the section regarding regulatory frameworks in four Nordic countries. in addition, Niko Karvosenoja has analysed the emission data for all the four cities. Anu Kousa and Jarkko V. Niemi have compiled the RWC
emission inventory in the Helsinki Metropolitan Area.

## 9. Competing interests

The authors declare that they have no conflict of interest.