# Peer review of "The influence of residential wood combustion on the concentrations of PM2.5 in four Nordic cities"

_Atmospheric Chemistry and Physics, 2019_

## Referee Comment (RC1) · Anonymous Referee #1 · 27 Aug 2019

Referee Comment ACP-2019-564

General Comment: The paper presents a detailed study of residential wood combustion as a source of PM2.5 in four different cities, one each in of four different Scandinavian countries. The study encompasses measurements, emissions inventories, and dispersion modeling for each city and surrounding area. The authors have done a credible job of pulling together a number of disparate sets of measurement, emissions, and modeling activities into a single manuscript. This is both a strength and a weakness of the paper, but in this referee's opinion, the balance is strongly on the "strength" side. Given the large variety of data sources, they did a fine job keeping focus and

consolidating where possible. I do have a few suggestions to improve the paper for the reader, which I will detail below. With the minor suggested improvements, I expect the manuscript will be publishable.

The major improvement I suggest is to tackle the numerous differences in the data sources in a much more head on fashion. The authors do a very good job of describing the differences in measurement methods in Section 2.1.2, the differences in emissions inventories in Section 2.2, and the differences in dispersion models in Section 2.3. What is missing in each case is a discussion of the significance of the differences. Are the differences important? Do they affect the results in any way? How might one evaluate the effects of these differences in the overall study. I could see a paragraph at the end of each section acknowledging the situation and evaluating its importance to the study and its results.

Specific Comments: Lines 442-446: This section could be described a little more clearly, and reasoning for the adjustment explained. Why did the measured background values need to be adjusted based on dispersion model calculations? What was the measured value before adjustment? How is the Bredkalen station considered background if its values need to be adjusted?

Lines 505-511: The authors state that the modeling system "has been evaluated", but give no indication of what the evaluation concluded. Was it determined to be accurate and appropriate for the study? Or were there problems? If there were problems discovered what were they and have they been addressed?

Line 520: Were chemical reactions and aerosol transformation processes included in the other dispersion modeling systems? If so, did they affect the results? What kind of impact might these processes have on the Helsinki results?

Lines 603-608: Figure 3 – I suggest the third set of panels for this figure. Add a top panel with the total PM2.5 for each of the sites organized by classification like the FB and IA panels. Line 650: The purple color for the highest emissions should be

changed. Maybe use a brown or a brown-red?

Lines 734-737: There is no mention of topography anywhere in the text. It looks to me like topography is a major factor in the high concentration areas of Oslo. I did not do an independent check, but am guessing that the high concentrations in the eastern section of the domain are in a valley where wood smoke (and other pollutants) tend to "pool" during nocturnal and other inversions.

Technical Corrections: Line 48: Add the word "the" before "EU". Line 75: Replace "contributed" with "attributed". Line 125: Replace "evaluated" with "determined". Line 686: Change "is" to "are".
* * *

---

## Referee Comment (RC2) · Anonymous Referee #2 · 4 Nov 2019

This manuscript estimates the impact of residential wood combustion (RWC) on PM2.5 over 4 Nordic cities using several data collection and modeling approaches. By integrating emissions data and modeling tools for each urban area, the authors characterize and compare the contribution of RWC to air pollution at different locations. This represents a significant undertaking to address an important research question. However, the manuscript's quality must be improved before being considered for publication. Specifically, a clearer description of methods and results must be provided, and the conclusions stated should be better supported. The limitations of the study's methods should be acknowledged and reported estimates better placed into context. Major and minor concerns associated with the current manuscript are listed below.

[Figure]

Major comments:

1) Use of English must be improved. The language used throughout the manuscript can be clearer and more concise. The overuse of passive voice often makes sentences confusing.

2) Figure quality should be improved. Figures lack labels and are low resolution (specific comments are listed below).

3) The introduction and literature review are disorganized. In the introduction, the authors fail to clearly convey the key points of the study and its motivation. Instead, the introduction jumps between locations, methods, and data sources. In its current form, it is difficult to understand the current state of knowledge on this topic and how this study fits into the larger body of research.

4) Several elements of the methodology are unclear. The intent of the study often gets lost in the language and details, some of which are unnecessary and do not address the more important aspects of the work. The manuscript is long and yet information important to understanding the work appears to be missing. Clarification and reorganization throughout the text should be considered to improve readability and highlight the study's main contributions.

5) As it stands now, the study largely focuses on the modeling approaches – descriptions of aspects of the different modeling setups for each urban area and their predictions are a large component of the manuscript. Nevertheless, as a modeling study, the analysis often can be superficial and incomplete. Conclusions are drawn from a patchwork of four different modeling frameworks. Still, the modeling systems are only briefly described and limitations/differences between them not clearly identified. Comparisons between them should be better consolidated and considered/discussed in the analyses of the modeled PM2.5 fields.

6) Inventory compilation is not discussed at depth. The methods used for RWC use

and emissions estimates can be unclear.

7) The RWC impacts on PM2.5 levels, the paper's main contribution, are not presented until page 24, after an overly verbose description of less relevant elements. The manuscript devotes a large amount of text to the total PM.5 modeled fields, rather than RWC impacts. Beyond justifying the use of these modeling systems to simulate atmospheric pollution at these locations, this seems to distract from the main objective of the manuscript. A clearer focus on the RWC-attributable PM2.5 concentrations and deeper analysis of these (e.g. a spatial map of the RWC PM2.5 instead the fractional contribution in fig. 7, a description of seasonal/monthly variability, and estimated of maximum daily impacts) would strengthen the manuscript.

8) A clearer explanation of why only some urban emissions sources are included in each urban area's simulation is required. Has previous work shown that the sources considered capture the majority of urban emissions and pollution? Additionally, it is unclear if the regional simulations used to estimate Helsinki, Oslo, and Copenhagen background concentrations include the urban area's emission from the urban areas themselves. If so, how does the study avoid double counting these emissions when finer-scale dispersion from urban sources is added to the regional background concentrations?

9) There is no mention of atmospheric chemistry, in particular, secondary aerosol formation. Prior research has shown that indoor combustion emissions can generate substantial secondary particulate matter, depending on combustion and atmospheric conditions. However, this is not discussed. While, the modeling tools applied at the urban scale do not include chemical or physical transformations, at a minimum the authors must discuss this limitation and how they would expect the inclusion of atmospheric chemistry to impact their results.

10) The conclusions section is mostly a repetition of the results. The authors should provide a deeper discussion of the significance of their RWC quantification and how it

fits into the large body of research, including insights, implications, and limitations of the work, as well as future research needs.

11) Policy is mentioned in the abstract (as a strongly worded recommendation) and briefly discussed in the introduction, yet this implication discussion was moved to the appendix. If this is an important implication, it should be discussed in the main body of the article and the authors should identify what specific policy implications the work has.

Additional comments:

- Line 24: The text "the higher long-range transported background" is unclear and should be reworded.

- Line 40: Little discussion or reasoning is provided to specifically support this statement: "Stricter and more efficient emission regulations should be set in the Nordic countries with respect to RWC".

- Line 45: Support this statement with references.

- Line 57-58: Remove this sentence: "Clearly, the burning of oil, coal . . . scope of RWC"

- Line 75: Throughout the manuscript, the word "contributed" is used incorrectly. The correct word here is "attributed".

- Line 82: Where were the Nordic observations reported?

- Line 105-107: This sentence is unclear and should be reworded.

- Line 110: Again "corresponded" is used incorrectly. Here, the intended meaning is "was responsible for".

- Line 115: Replace "ca." with "approximately". I would also recommend avoiding "viz."

- Line 127: Replace with "colder half of the year"

- Line 127-129: This sentence is poorly written.

- Figure 2: The figure is low resolution and low quality. Making colors, labels and features uniform among panels would help. Panels are not labeled.

- Section 2.1.1: This section is not very informative with respect to main findings, and the information within could be moved to the Appendix to improve conciseness

- Line 241: The term "addressed" appears to be used incorrectly.

- Line 245: What do the authors mean by an "urban background site"?

- Line 257: What criteria were used to determine that these 3 stations are RWC influenced (and the others not)?

- Line 286 and 288: Again "assessment" appears to be used incorrectly.

- Line 298-299: Cite the prior work on measurements and emission factors that this mentioned.

- Line 300-305: Why is temperature and resolution discussed in this paragraph, is a connection trying to be made?

- Line 347-349: Better describe how the different data sources were combined.

- Line 354: Here and throughout the manuscript, the word "evaluated" (similar to "assessed") could be replaced with "estimated" to add clarity.

- Line 439: How were the background concentrations "slightly adjusted"?

- Lines 442-446: The adjustment applied and its justification are unclear.

- Line 476-479: The description of the use of the OSPM model is unclear.

- Line 530-536: Were meteorological fields interpolated? If so, how?

- Line 560-562: Are urban Copenhagen emissions, in particular, RWC emissions, not included in the emissions inventory? If so, how is double counting these avoided?

- Section 2.4: Why were these 2 metrics selected for evaluation?

- Figure 3: Markers are inconsistent, panels are not labeled.

- Line 609-610: This sentence is unclear.

- Line 619-621: The authors should discuss what they believe is the reason for inferior performance at these stations.

- Line 625-628: Is there specific evidence supporting the statement about the resolution being responsible for lower IA?

- Figure 4: The quality of the figure can be improved and reorganized to more easily compare across cities. In the caption, change "long-term" to "annual".

- Figure 5: Label panels. The color scale is misleading, with some intervals representing a 0.02 ton difference and others 1 ton. This makes interpretation difficult.

- Lines 657-660: This paragraph appears to be unnecessary.

- Line 661: What do the authors mean by "spatially averaged maximum emission values"?

- Line 665: Does this mean the highest emissions within Helsinki and Copenhagen, or across the 4 urban areas?

- Line 712-713: This sentence is unclear: "The concentration distributions from RWC . . ."

- Figure 6: Labels are missing on panels. On the color scale, why are some intervals 1 ug/m3 and others 0.5 ug/m3? As mentioned in the major comments, maps of RWC-derived PM2.5 would be more informative than these for total PM2.5.

- Line 740-741: Support this statement with references.

- Figure 7: Panels should be labeled. Again, the color bar is misleading by using varying interval sizes.

- Line 773: Define "spatial ranges"

- Figure 8: The quality of this figure would be greatly improved by using a box plot (or another visualization of distributions) instead of the overly simple bar plot included. Titles on axes are unclear.

- Line 795: The second reason mentioned does not appear to be identified.

- Line 862: Why are annual average values reported instead of seasonal values if RWC appears to be largely concentrated into colder seasons?

---

## Author Comment (AC1) · 8 Jan 2020

**Response to reviewers' comments, 8 January 2020**

For clarity, in the following the reviewers' comments have been indicated in blue font. Our responses to reviewers' questions are in black font, and our descriptions on the changes made to the revised manuscript are written in red font.

**Anonymous Referee #1**

Referee Comment ACP-2019-564

General Comment: The paper presents a detailed study of residential wood combustion as a source of PM2.5 in four different cities, one each in of four different Scandinavian countries. The study encompasses measurements, emissions inventories, and dispersion modeling for each city and surrounding area. The authors have done a credible job of pulling together a number of disparate sets of measurement, emissions, and modeling activities into a single manuscript. This is both a strength and a weakness of the paper, but in this referee's opinion, the balance is strongly on the "strength" side. Given the large variety of data sources, they did a fine job keeping focus and consolidating where possible. I do have a few suggestions to improve the paper for the reader, which I will detail below. With the minor suggested improvements, I expect the manuscript will be publishable.

Thank you for the favourable overall review.

The major improvement I suggest is to tackle the numerous differences in the data sources in a much more head on fashion. The authors do a very good job of describing the differences in measurement methods in Section 2.1.2, the differences in emissions inventories in Section 2.2, and the differences in dispersion models in Section 2.3. What is missing in each case is a discussion of the significance of the differences. Are the differences important? Do they affect the results in any way? How might one evaluate the effects of these differences in the overall study. I could see a paragraph at the end of each section acknowledging the situation and evaluating its importance to the study and its results.

We have added the sections that were suggested by the referee, at the end of the sections 2.1.2 (measurement networks), 2.2 (emission inventories), and 2.3. (dispersion modelling). We have tried our best to evaluate the significances of the methodical differences and their influence on the overall results and conclusions of this study.

We have added a new section at the end of 2.1.2, titled as "Inter-comparison of the measurement networks in the target cities", at the end of section 2.2., called 2.2.3, titled as "Inter-comparison of the emission inventories in the target cities", and at the end of section 2.3., called "2.3.3. Inter-comparison of the dispersion modelling in the target cities".

Specific Comments: Lines 442-446: This section could be described a little more clearly, and reasoning for the adjustment explained. Why did the measured background values need to be

adjusted based on dispersion model calculations? What was the measured value before adjustment? How is the Bredkalen station considered background if its values need to be adjusted?

The available regional background measurement stations are located at substantial distances from Umeå, especially the Bredkälen station. In addition, dispersion modelling results show that the regional background exactly at the Bredkälen location is somewhat lower than the regional background at the Umeå location (which is about 4.0 $\mu gm^{-3}$). We therefore obtained a more accurate estimate of the regional background by combining measured data and the results of dispersion model computations.

The text in section 2.3. (Atmospheric dispersion modelling for Umeå) has been clarified. Some repetition was also removed from this description.

Lines 505-511: The authors state that the modeling system "has been evaluated", but give no indication of what the evaluation concluded. Was it determined to be accurate and appropriate for the study? Or were there problems? If there were problems discovered what were they and have they been addressed?

We have added clarification of the results of model evaluation. However, an in-depth analysis of the advantages and challenges of the various models are outside the scope of this study.

Statement on the overall CAR-FMI model performance for $PM_{2.5}$ was added to the text in section 2.3.2, including example results from an extensive model evaluation for London.

Line 520: Were chemical reactions and aerosol transformation processes included in the other dispersion modeling systems? If so, did they affect the results? What kind of impact might these processes have on the Helsinki results?

Chemical reactions were included in the regional scale computations, for all the cities. However, aerosol transformation (i.e., nucleation, condensation and evaporation, coagulation, deposition, etc.) on urban scale was not specifically included for any of the cities. This could be a topic of a separate study. In case of urban computations, gas-to-particle transformations are usually fairly slow reactions that do not have a major influence on urban distance scales in Nordic cities.

Karl et al. (2016) evaluated the impacts on particulate number concentrations and particle number size distributions of including aerosol processes on local and urban scales, including modelling of measurement campaigns in Oslo and Helsinki (Ref: Karl, M., Kukkonen, J., Keuken, M. P., Lützenkirchen, S., Pirjola, L., and Hussein, T., 2016. Modeling and measurements of urban aerosol processes on the neighborhood scale in Rotterdam, Oslo and Helsinki. Atmos. Chem. Phys., 16, 4817-4835, doi:10.5194/acp-16-4817-2016, 2016. http://www.atmos-chem-phys.net/16/4817/2016/). These results are partly relevant also for $PM_{2.5}$, regarding the results for the accumulation mode (as most of the mass of $PM_{2.5}$ is in this mode).

Based on Karl et al. (2016) we can conclude that the impact of aerosol processes on the annually averaged $PM_{2.5}$ concentrations would be minor (smaller than the corresponding impact of other uncertainties in, e.g., the emission inventories). However, their impact in specific conditions, such as air quality episodes, and the impact on smaller than accumulation mode particles, could be substantial. However, this study addressed mainly annually averaged $PM_{2.5}$ concentrations.

The dry deposition of PM$_{2.5}$ on urban scale can be allowed for in the CAR-FMI model (i.e., for the computations in Helsinki); however, it has previously been shown that this effect is of minor importance on an urban scale.

We have added a clarification on the modelling of chemistry and aerosol processes, and their importance to the section 2.3.1 "Overview of dispersion modelling" (second paragraph).

Lines 603-608: Figure 3 – I suggest the third set of panels for this figure. Add a top panel with the total PM2.5 for each of the sites organized by classification like the FB and IA panels.

We agree that it will be useful to present the average (total) PM$_{2.5}$ at the urban and regional stations, for each city. However, these have been presented in Fig. 4, both for modelled and measured values.

The data presented in Fig. 4 has been re-structured, to allow an easier inter-comparison of measured and predicted results (i.e., presenting these side by side), for each city, and for the urban and regional background stations, respectively.

Line 650: The purple color for the highest emissions should be changed. Maybe use a brown or a brown-red?

We have re-drawn Figs. 5a-d (the spatial distributions of emissions), as well as other map-based results, Figs 6 and 7. The revised Figs. 5a-d are much easier to comprehend and will help the readers to understand better the emission inventories from RWC.

We have used only constant intervals in the legends of Figs. 5-7, as per reviewer 2 request. We have also selected the color scales in a more informative manner, i.e., so that the lower values are represented by lighter colours, and the highest concentrations by bright colours. The highest values are represented as black, dark violet and red in Figs 5a-d.

Lines 734-737: There is no mention of topography anywhere in the text. It looks to me like topography is a major factor in the high concentration areas of Oslo. I did not do an independent check, but am guessing that the high concentrations in the eastern section of the domain are in a valley where wood smoke (and other pollutants) tend to "pool" during nocturnal and other inversions.

Yes, topography is certainly one of the factors that influence air pollution in Oslo. We have written in the manuscript (in Appendix A in the revised manuscript): "The city of Oslo and the Greater Oslo Region are situated at the northernmost end of a fjord and surrounded by hills that have heights of approximately 500 m above the sea level."

Topography is included as input data in the regional scale modelling for Oslo domain. The topography within the domain is defined on the Eulerian grid in terms of the elevation above sea level. A more detailed description is in the article by Hamer et al. (2019), https://doi.org/10.5194/gmd-2019-199.

The high concentrations in the eastern part of the domain were caused by several factors. We have assessed the largest factor to be the intensive residential wood combustion activity in this region. Topography has also influence, as this region is at lower ground than the surrounding hills. However, the topography of this region is not substantially different from most other regions within this domain. It can also be concluded based on Fig. 5c that the road and street network is fairly dense in this region; the vehicular emission is also a contributing factor.

Within the section "Atmospheric dispersion modelling for Oslo", we have added description on how topography was included in the modelling (second to last paragraph).

We have added a discussion of the potential influence of topography for the high contributions from RWC in Oslo, to the section 3.3 "PM2.5 concentrations and source contributions from RWC".

Technical Corrections: Line 48: Add the word "the" before "EU". Line 75: Replace "contributed" with "attributed". Line 125: Replace "evaluated" with "determined".

The suggested corrections and changes were made in the manuscript.

Line 686: Change "is" to "are".

Corrected.

**Anonymous Referee #2**

This manuscript estimates the impact of residential wood combustion (RWC)onPM2.5 over 4 Nordic cities using several data collection and modeling approaches. By integrating emissions data and modeling tools for each urban area, the authors characterize and compare the contribution of RWC to air pollution at different locations. This represents a significant undertaking to address an important research question. However, the manuscript's quality must be improved before being considered for publication. Specifically, a clearer description of methods and results must be provided, and the conclusions stated should be better supported. The limitations of the study's methods should be acknowledged and reported estimates better placed into context. Major and minor concerns associated with the current manuscript are listed below.

Major comments:

1) Use of English must be improved. The language used throughout the manuscript can be clearer and more concise. The overuse of passive voice often makes sentences confusing.

We have checked the language throughout the manuscript. The have reduced the use of passive voice.

We have tried to improve the language in numerous instances. E.g., in abstract, both plural first person and passive are used. Also in 'aims of the study' (last paragraph in introduction), both plural first person and passive are used. We have used British spelling and checked the spelling in words that have a different form in American English, such as, e.g., 'metre', 'centre' and 'behaviour'.

2) Figure quality should be improved. Figures lack labels and are low resolution (specific comments are listed below).

We agree with the referee that these can be improved.

We have re-drawn all figures from Fig 2 to Fig.8 (details are listed below).

Labels have been added to all sets of figures. We have used only constant intervals in the legends of Figs. 5-7. We have also selected the color scales in a more informative manner, i.e., so that the lower values are represented by lighter colours, and the highest concentrations by bright colours.

3) The introduction and literature review are disorganized. In the introduction, the authors fail to clearly convey the key points of the study and its motivation. Instead, the introduction jumps between locations, methods, and data sources. In its current form, it is difficult to understand the current state of knowledge on this topic and how this study fits into the larger body of research.

We have re-written a substantial part of the introduction. The revised introduction includes a much better motivation of the study.

The revised literature review has been structured so that it first addresses global studies, then regional ones (especially regarding Europe), and finally local ones (with focus on the Nordic

region). We felt that this is the most logical and clear procedure for writing the literature review. The whole introduction has been checked and restructured to be in accordance to the above principle.

As some of the referenced studies are experimental, some modelling (both using a wide variety of methods), and some both of these, we did not find any single ideal way to classify the literature. E.g., if we would have decided presenting first the modelling studies, then experimental studies, this would result in jumping from global to local and back.

In the beginning of the introduction (first and third paragraphs in the revised manuscript), we have added motivation for this study, including a range of new citations. These comments and references describe much better the health impacts caused specifically by RWC; the health impacts are the main motivation for this study. We refer also to the health impacts in the developing countries, to the wide range of pollutants that can be produced by RWC, and to the various health outcomes that may be attributed to RWC.

We have also moved the paragraph that addresses economic effects within some continents and some countries (which starts "Brandt et al. (2013)" to a more logical place, after the global studies. Also all the parts of the introduction that addressed primarily health impacts were moved to the proper place in the first part of the introduction (third paragraph).

We have added two citations and some description on two recently published relevant studies, on the significance of RWC in Nordic countries, Im et al. (2019), and on the emission modelling and seasonal variation of RWC by Grythe et al. (2019).

4) Several elements of the methodology are unclear. The intent of the study often gets lost in the language and details, some of which are unnecessary and do not address the more important aspects of the work. The manuscript is long and yet information important to understanding the work appears to be missing. Clarification and reorganization throughout the text should be considered to improve readability and highlight the study's main contributions.

We have improved the clarity of the text at several instances, and added summarizing texts for easier reading (e.g., please see a more detailed description in our response to referee 1's major comments). Most of the figures have been re-drawn to be more easily understandable. The conclusions section has been re-written. We have also done numerous other improvements to the text, as specified below.

5) As it stands now, the study largely focuses on the modeling approaches – descriptions of aspects of the different modeling setups for each urban area and their predictions are a large component of the manuscript. Nevertheless, as a modeling study, the analysis often can be superficial and incomplete. Conclusions are drawn from a patchwork of four different modeling frameworks. Still, the modeling systems are only briefly described and limitations/differences between them not clearly identified. Comparisons between them should be better consolidated and considered/discussed in the analyses of the modeled PM2.5 fields.

We have added more description and analysis on the differences and similarities of the modelling systems and experimental methods, and evaluation on how these differences will affect the final results and conclusions. This was also asked by referee 1. Please find a more detailed description in our response to referee 1's major comments, and in our responses to several other referee comments.

We have added a new section at the end of 2.1.2, titled as "Inter-comparison of the measurement networks in the target cities", at the end of section 2.2., called 2.2.3, titled as "Inter-comparison of the emission inventories in the target cities", and at the end of section 2.3., called "2.3.3. Inter-comparison of the dispersion modelling in the target cities".

6) Inventory compilation is not discussed at depth. The methods used for RWC use and emissions estimates can be unclear.

The evaluation of the emissions from RWC is a complex and challenging task. For one city only, collecting such an inventory may require at least one man-year of work, or more. Such a work relies on various register datasets and surveys, and will involve also some modelling. The reviewer may possibly refer in his/her comment on the nature and details of these underlying datasets and surveys. However, we felt that it is not possible to present these in detail in a journal article. We therefore had to rely on citing the references of many of these datasets, emission coefficient evaluations, models, etc.

However, we have tried to include a fairly detailed description especially on the emission inventories for RWC. These are presented in section 2.2. on more than 4 pages, and emission inventories for other source categories are additionally presented in Appendix A, on approximately two pages.

We have added more description and analysis on the differences and similarities of the emission inventory methods, and evaluation on how these differences will affect the final results and conclusions.

We have added a new section at the end of section 2.2., called 2.2.3, titled as "Inter-comparison of the emission inventories in the target cities".

We have also re-drawn the Figs. 5a-d (the spatial distributions of emissions). In our view, the revised figure is much easier to comprehend, and will help the readers to understand better the emission inventories from RWC. We have used only constant intervals in the legend, as per reviewer request. We have also selected the color scales in a more informative manner, i.e., so that the lower values are represented by much lighter colours (light green and light yellow), and the highest concentrations by bright colours (black, violet and red).

7) The RWC impacts on PM2.5 levels, the paper's main contribution, are not presented until page 24, after an overly verbose description of less relevant elements. The manuscript devotes a large amount of text to the total PM.5 modeled fields, rather than RWC impacts. Beyond justifying the use of these modeling systems to simulate atmospheric pollution at these locations, this seems to distract from the main objective of the manuscript. A clearer focus on the RWC-attributable PM2.5 concentrations and deeper analysis of these (e.g. a spatial map of the RWC PM2.5 instead the fractional contribution in fig. 7, a description of seasonal/monthly variability, and estimated of maximum daily impacts) would strengthen the manuscript.

For practical purposes, e.g., for finding the most cost-efficient ways to reduce pollution, it is crucial to know, what is the fraction of RWC attributed pollution in relation to pollution from other sources. We have therefore tried to compile as complete emission inventories as possible and evaluate also the total concentrations attributed to all the relevant source categories. This will help the reader to set the contribution of pollution from RWC into a proper context.

Our focus has been on RWC attributed pollution. For instance, we have addressed in relatively more detail the RWC emission inventories. The emission inventories from all the other sources were addressed much more briefly.

Regarding the spatial maps and presenting the contributions from RWC both as absolute concentrations and as percentages, we have substantially improved the presentations, as per reviewer requests.

We have selected to present the spatial distributions of the total concentrations and the corresponding RWC fractions. Multiplying these two values will produce the contributions from RWC as absolute concentration values. Representing also the spatial distributions of the RWC as absolute concentrations would therefore in our view be redundant. However, in the spirit of the referee's comment, we have separately presented the ranges of the RWC contributions, both as absolute concentrations (as asked for by the reviewer) and as percentages in Fig. 8.

Presenting and analyzing the seasonal or monthly variation of RWC emissions or concentrations is outside the scope of this study. The seasonal variation of RWC emissions has been evaluated, e.g., by Grythe et al. (2019). They show the seasonality of RWC and diurnal time variation based on observed and modelled concentration at different Norwegian urban air quality stations. However, we added a brief description and citation to this study to the manuscript (to the introduction).

Grythe, Henrik, Susana Lopez-Aparicio, Matthias Vogt, Dam Vo Thanh, Claudia Hak, Anne Karine Halse, Paul Hamer,and Gabriela Sousa Santos, 2019. The MetVed model: development and evaluation of emissions from residential wood combustion at high spatio-temporal resolution in Norway. Atmos. Chem. Phys., 19, 10217–10237, 2019 https://doi.org/10.5194/acp-19-10217-2019.

We have re-drawn Figs. 6a-d (the spatial distributions of total concentrations), Figs. 7a-d (the spatial distributions of source contributions of RWC) and Fig. 8 (the ranges of RCW contributions as absolute concentrations and percentages). The revised Figs. 7a-d and 8a-d are much easier to comprehend, and will help the readers to understand better these results. We have used only constant intervals in these legends, as per reviewer request. We have also selected the color scales in a more informative manner, i.e., so that the lower values are represented by lighter colours and the highest concentrations by brighter colours.

Regarding Fig. 8, the texts on the vertical axes have been improved to be more accurate and better representative, as per reviewer request.

8) A clearer explanation of why only some urban emissions sources are included in each urban area's simulation is required. Has previous work shown that the sources considered capture the majority of urban emissions and pollution?

Yes, the reviewer quessed correctly: previous studies have shown that the included sources capture the vast majority of $PM_{2.5}$ pollution within these regions. We have added more description and analysis on these matters. This was also asked by referee 1.

We have added a new section at the end of section 2.2., called 2.2.3, titled as "Inter-comparison of the emission inventories in the target cities".

Additionally, it is unclear if the regional simulations used to estimate Helsinki, Oslo, and Copenhagen background concentrations include the urban area's emission from the urban areas themselves. If so, how does the study avoid double counting these emissions when finer-scale dispersion from urban sources is added to the regional background concentrations?

In all of these simulations, we have selected the regional background to be evaluated at a regional background measurement site. These measurement sites have in previous studies been evaluated to be affected only by very small percentages by the transport of the local urban emissions.

9) There is no mention of atmospheric chemistry, in particular, secondary aerosol formation. Prior research has shown that indoor combustion emissions can generate substantial secondary particulate matter, depending on combustion and atmospheric conditions. However, this is not discussed. While, the modeling tools applied at the urban scale do not include chemical or physical transformations, at a minimum the authors must discuss this limitation and how they would expect the inclusion of atmospheric chemistry to impact their results.

Could you please read our reply for referee 1, his/her question "line 520; Were chemical reactions …". These questions by referee 1 and 2 were very similar.

10) The conclusions section is mostly a repetition of the results. The authors should provide a deeper discussion of the significance of their RWC quantification and how it fits into the large body of research, including insights, implications, and limitations of the work, as well as future research needs.

We have re-written the conclusions section, as per reviewer suggestions.

We have added to the beginning of the conclusions (two first paragraphs) a description on how this study fits into the larger body of research. This discussion also indicates the significance of this study, as the first harmonized assessment of the role of RWC in the Nordic region.

We have also added a description in the revised conclusions (third and fourth paragraphs), which addresses the limitations and challenges of this work.

The discussion of the main results and conclusions of this study was also revised and shortened. The main results were addressed on a more general level (e.g., avoiding excessive numerical values on specific computations).

We have also added a paragraph on the future research needs (second to the last paragraph).

11) Policy is mentioned in the abstract (as a strongly worded recommendation) and briefly discussed in the introduction, yet this implication discussion was moved to the appendix. If this is an important implication, it should be discussed in the main body of the article and the authors should identify what specific policy implications the work has.

Detailed recommendations on policy are not within the main scope of the paper; policy aspects are therefore presented in an Appendix.

We have removed the comments on policy from the abstract.

Additional comments:

- Line 24: The text "the higher long-range transported background" is unclear and should be reworded.

This has been reworded.

- Line 40: Little discussion or reasoning is provided to specifically support this statement: "Stricter and more efficient emission regulations should be set in the Nordic countries with respect to RWC".

This statement was removed.

 - Line 45: Support this statement with references.

Three references from Patel et al., 2013; Sigsgaard et al., 2015, and Butt et al., 2016 were added in the sentence to support the statement.

-Line57-58: Remove this sentence: "Clearly, the burning of oil, coal ... scope of RWC"

This sentence was removed.

- Line 75: Throughout the manuscript, the word "contributed" is used incorrectly. The correct word here is "attributed".

We have revised the manuscript and replaced the word 'contributed' to with 'attributed' where appropriate.

- Line 82: Where were the Nordic observations reported?

Corrected. Sorry, this has to read 'Norwegian' instead of 'Nordic'.

- Line 105-107: This sentence is unclear and should be reworded.

Corrected.

- Line 110: Again "corresponded" is used incorrectly. Here, the intended meaning is "was responsible for".

We changed 'corresponded' to 'was responsible for' per Reviewer's#2 request.

- Line 115: Replace "ca." with "approximately". I would also recommend avoiding "viz."

We replaced "ca." with "approximately" and 'viz.' with phrases such as 'more specifically', 'in particular', 'for example' and 'such as' as the reviewer suggested. In one case, and in particular, in line 582, 'viz.' was removed.

- Line 127: Replace with "colder half of the year"

We replaced the phrase 'colder half-year' with 'colder half of the year' as Reviewer #2 suggested.

- Line 127-129: This sentence is poorly written.

Corrected

- Figure 2: The figure is low resolution and low quality. Making colors, labels and features uniform among panels would help. Panels are not labeled.

Figure 2 has been re-drawn, and it is now presented using an improved resolution and quality. Panels are now labeled appropriately, and both features and colors are uniform as per reviewer 2 request. Information on the stations from each location has also been added to the caption of Fig. 2.

- Section 2.1.1: This section is not very informative with respect to main findings, and the information within could be moved to the Appendix to improve conciseness

We have moved this section to an Appendix (Appendix A in the revised manuscript).

- Line 241: The term "addressed" appears to be used incorrectly.

Corrected.

- Line 245: What do the authors mean by an "urban background site"?

Urban background site refers to a location that is representative for the exposure of a majority of population in a city. We have in this study simply used the classifications evaluated by the local environmental councils in each city, according to the guidance by the EU.

Urban background stations can be located, e.g., in parks and other locations that are likely to be more representative of general city-wide exposure. It is also possible to place urban background stations in suburban areas, to monitor exposure in areas less dominated by traffic. The urban background may be influenced by traffic emissions in very different degrees across Europe, depending on the regional background concentrations, urban topography, urban density, structure of the urban road network, traffic density and fleet composition.

- Line 257: What criteria were used to determine that these 3 stations are RWC influenced (and the others not)?

These stations are located in residential regions, at places with less direct influence from vehicular, shipping or industrial sources. The housing in these areas also contain mostly detached houses, instead of large blocks of flats.

- Line 286 and 288: Again "assessment" appears to be used incorrectly.

In line 286 'assessment' was replaced with 'inventory', while in line 288 we deleted it.

- Line 298-299: Cite the prior work on measurements and emission factors that this mentioned.

These have been cited in Table 1. We have organized this section so that the detailed citations for each city are in table, and the text is more general.

- Line 300-305: Why is temperature and resolution discussed in this paragraph, is a connection trying to be made?

We have divided this paragraph into two paragraphs and written this text section more clearly. Our intention was not to combine temperature dependency and spatial resolution.

- Line 347-349: Better describe how the different data sources were combined.

We have written this combination in more detail and more clearly.

- Line 354: Here and throughout the manuscript, the word "evaluated" (similar to "assessed") could be replaced with "estimated" to add clarity.

In this instance, we replaced 'evaluated' with 'estimated'. We have also checked and revised the language throughout the manuscript.

- Line 439: How were the background concentrations "slightly adjusted"?

This has been written more clearly in the first paragraph of the section "Atmospheric dispersion modelling for Umeå".

- Lines 442-446: The adjustment applied and its justification are unclear.

This has been written more clearly in the first paragraph of the section "Atmospheric dispersion modelling for Umeå".

- Line 476-479: The description of the use of the OSPM model is unclear.

This paragraph has been re-written, to be more clear.

- Line 530-536: Were meteorological fields interpolated? If so, how?

The variables related to wind and atmospheric stability were used as input in a preprocessing diagnostic wind field model. It was therefore not necessary to numerically interpolate the meteorological data; instead the diagnostic wind model will assimilate the met observations.

- Line 560-562: Are urban Copenhagen emissions, in particular, RWC emissions, not included in the emissions inventory? If so, how is double counting these avoided?

The issue of possible double counting has been taken into account in the coupling of the models DEHM and UBM. All Danish emissions are included in both the DEHM model and the UBM model, however, they operate with different resolutions. The DEHM model provided the regional background concentrations 25 km upstream from all the individual gridded receptor points in the UBM model, and the UBM model was subsequently used for calculating the local scale contribution from local emissions up to 25 km from the receptor points. The upstream concentrations (as provided by the DEHM model) were computed separately for each hour of the year in the upwind direction with respect to the smaller domain, depending on the wind direction.

In this way, double counting of local emissions is avoided. In the UBM model run in this study, we have calculated the local RWC contribution from emissions within the smaller model domain, which includes Copenhagen and its surroundings (eastern part of Zeeland).

- Section 2.4: Why were these 2 metrics selected for evaluation?

We have added a clarification: "The IA is a measure of the agreement of the measured and predicted timeseries of concentrations, and the FB is a measure of the agreement of the longer term (e.g., annual) average concentrations." In other words, we have selected one parameter for the correlation of hourly or daily values, and another for the agreement of the overall (annual) averages.

- Figure 3: Markers are inconsistent, panels are not labeled.

We revised the markers and added labels.

- Line 609-610: This sentence is unclear.

The whole paragraph has been written more clearly.

- Line 619-621: The authors should discuss what they believe is the reason for inferior performance at these stations.

We have added an explanation of the worst under-predictions. The detailed reasons for the under- or over-predictions for the other mentioned stations are not known with certainty; in general, these reflect the uncertainties of modelling urban concentrations. However, it is well-known from previous research that urban background models, such as the one applied for Copenhagen in this study, will underestimate the concentrations at traffic sites.

-Line625-628: Is there specific evidence supporting the statement about the resolution being responsible for lower IA?

Yes, there is. We wrote in the manuscript: "In particular, the IA values for the traffic stations are lower for Copenhagen, compared with the corresponding values in Helsinki and Oslo. This is due to the coarser spatial model resolution (1 x 1 km$^2$) in Copenhagen, compared with those in the other three target cities, which tends to result in an underprediction of the local influence of vehicular traffic. A better model performance was obtained in a previous study for the street stations in Copenhagen, when the street pollution model OSPM was used (Khan et al., 2018)."

In other words, using additionally a street canyon dispersion model, one will achieve a better agreement. It is also known from several other previous studies that the 1 km resolution is not ideal, in case of stations located within or very near major streets. However, the main focus of this study was on RWC pollution, instead of vehicular pollution.

- Figure 4: The quality of the figure can be improved and reorganized to more easily compare across cities. In the caption, change "long-term" to "annual".

We have re-structured the order of the columns; the new version allows an easier comparison of measured and predicted values. In our view, the comparison across cities is also easier. We also improved the technical quality of the figure. We also replaced 'long-term' with 'annual'.

- Figure 5: Label panels. The color scale is misleading, with some intervals representing a 0.02 ton difference and others 1 ton. This makes interpretation difficult.

Figure 5 has been re-drawn. Labels have been added, together with the city names for easier reading. All the intervals are now equal. The selection of the colour scale is also better.

- Lines 657-660: This paragraph appears to be unnecessary.

We deleted this paragraph.

- Line 661: What do the authors mean by "spatially averaged maximum emission values"?

This wording was unnecessarily technical, we have revised it.

- Line 665: Does this mean the highest emissions within Helsinki and Copenhagen, or across the 4 urban areas?

This means 'in the case of Helsinki and Copenhagen'. We have clarified the sentence.

- Line 712-713: This sentence is unclear: "The concentration distributions from RWC ..."

This has been clarified.

- Figure 6: Labels are missing on panels. On the color scale, why are some intervals 1 ug/m3 and others 0.5 ug/m3? As mentioned in the major comments, maps of RWC derived PM2.5 would be more informative than these for total PM2.5.

Figure 6 has been re-drawn. Labels have been added, together with the city names for easier reading. All the intervals are now equal. The selection of the colour scale is also better.

- Line 740-741: Support this statement with references.

Relevant references have been added.

- Figure 7: Panels should be labeled. Again, the color bar is misleading by using varying interval sizes.

Figure 7 has been re-drawn. Labels have been added, together with the city names for easier reading. All the intervals are now equal. The selection of the colour scale is also better in our view.

- Line 773: Define "spatial ranges"

We have removed 'spatial', as that may be misleading. This sentence refers to the range of annual average values within each domain. Figure 8 has been re-drawn to be more clear.

- Figure 8: The quality of this figure would be greatly improved by using a box plot (or another visualization of distributions) instead of the overly simple bar plot included. Titles on axes are unclear.

We revised the titles of the axes to be more clear.

Presenting the distributions of the two quantities presented in the figure would be possible, but that would unfortunately result in a much more complex presentation, including a series of 4 + 4 distribution panels. The aim of this figure is simply to illustrate the ranges of the RWC contributions.

- Line 795: The second reason mentioned does not appear to be identified.

This is more clear now, after revising Fig. 8.

-Line862: Why are annual average values reported instead of seasonal values if RWC appears to be largely concentrated into colder seasons?

The main reason for this was simplicity, and the need not to increase further the length of the manuscript. The duration of the so-called cold season of the year at each location varies from year to year, and this season is not necessarily the same for all the considered cities for any selected year. In addition, RWC is also used during the warmer season of the year, although the combusted amounts are smaller.

---

## Author Response (AR2)

**Response to editor's comments**

**1. Editor's comments**

Editor Decision: Publish subject to minor revisions (review by editor) (16 Feb 2020) by Anne Perring
Comments to the Author:
Dear Jakko and coauthors,

I appreciate the substantial revisions you have made in response to the reviewers comments. I have two additional/correlated comments of my own that I think would further strengthen the manuscript. Once these are addressed I will be happy to accept the paper for publication in ACP.

1) Reviewer 2 mentioned a couple of parts of the manuscript which struck them as long yet, in the revisions, little was cut while many things were added. The introduction especially strikes me as overly detailed and I would recommend rewriting to include only the background information necessary for the reader to follow the work at hand. As written it is a very thorough literature review with one to two sentences on each of many references. I would recommend reorganizing so that you have three or four thematic paragraphs in which you discuss, for example, 1) previous studies that have looked at health effects of RWC aerosol, 2) prior findings of fractional contribution of RWC sources in Europe, 3) findings about RWC in other places (if necessary) and so on.

2) Figure 4 does not need to be 3d or shown with perspective. I recommend revising to be a normal bar graph.

**2. Response to editor's comments**

We have shortened the introduction and reorganized the literature review material, as per the editor's suggestions. In our view, the new version of the introduction is more to the point, and more clearly structured.

Figure 4 has been presented two-dimensionally.